# Temporal evolution of flow-like landslide hazard for a road infrastructure in the Municipality of Nocera Inferiore (southerm Italy) under the effect of climate change

Marco Uzielli[1,2], Guido Rianna[3], Fabio Ciervo[3], Paola Mercogliano[3,4], Unni K. Eidsvig[2]

[1] Georisk Engineering S.r.l., Firenze, 50132, Italy
[2] NGI (Norwegian Geotechnical Institute), Oslo, 0855, Norway
[3] CMCC Foundation (Centro Euro-Mediterraneo sui Cambiamenti Climatici), Capua, 81043, Italy
[4] CIRA (Centro Italiano Ricerche Aerospaziali), Capua, 81043, Italy

*Correspondence to*: Marco Uzielli (muz@georisk.eu)

**Abstract.** In recent years, flow-like landslides have extensively affected pyroclastic covers of the Campania Region in southern Italy, causing victims and conspicuous economic damages. Due to the high criticality of the area, a proper assessment of future variations in event occurrences due to expected climate changes is crucial. The study assesses the temporal variation in flow-like landslide hazard for a section of the Autostrada A3 "Salerno-Napoli" motorway, which runs across the toe of the Monte Albino relief in the Nocera Inferiore municipality. Hazard is estimated spatially depending on: (1) the likelihood of rainfall-induced event occurrence within the study area; and: (2) the probability that the any specific location in the study area will be affected during the runout. The probability of occurrence of an event is calculated through the application of Bayesian theory. Temporal variations due to climate change are estimated up to 2100 through the EURO-CORDEX ensemble of climate simulations, accounting for current uncertainties in characterization of variations in rainfall patterns. Reach probability, defining the probability that a given spatial location is affected by flow-like landslides, is calculated spatially through based distributed empirical model. The outputs of the study predict substantial increases in occurrence probability over time for two different scenarios of future socio-economic growth and atmospheric concentration of greenhouse gases.

## 1 Introduction

In recent years, eminent scholars have debated about the main features of "shallow" and "deep" uncertainties in assessment of natural hazards (Stein & Stein, 2013; Hallegatte et al. 2012; Cox, 2012). Shallow uncertainties are associated to reasonably well know probabilities of outcomes (Stein & Stein, 2012), while deep uncertainties are associated to: (1) several possible future worlds without known relative probabilities; (2) multiple conflicting but equally-reasonable world-views; and (3) adaptation strategies with remarkable feedbacks among the sectors (Hallegatte et al. 2012).

As stressed in these works, climate change and its impacts can be considered "a fantastic example of 'very deep' uncertainty". Given the extent of potential impacts on communities (Paris Agreement, 2015) including their economic dimension (Stern, 2006; Nordhaus, 2007; Chancel & Piketty, 2015), considerable efforts have been made in recent years to assess the variations

in frequency and magnitude of weather-induced hazards in a changing climate (Seneviratne et al., 2012). A variety of strategies have been devised and implemented with the aim of detecting the main sources of uncertainty and their extent (Wilby & Dessai 2010; Cooke 2014; Koutsoyiannis & Montanari 2012; Beven 2015).

Investigations on future trends in the occurrence and consequences of weather-induced slope movements and on the uncertainties in their estimation have received relatively limited interest (Gariano & Guzzetti 2016; Beven et al. 2015). The paucity of investigations could be due to the mismatch between the usual scale of analysis for landslide case studies and the much coarser currently available horizontal resolutions of climate projections, as well as to the difficulty in generalizing findings to other contexts given the relevance of site-specific geomorphological features.

## 1.1 Previous studies of flow-like movements in pyroclastic soils in Campania

In the attempt to address the above limitations, several recent studies have focused on future variations in the occurrence of flow-like landslides or more in general flow-like movements affecting pyroclastic covers mantling the carbonate bedrocks in the Campania Region in southern Italy. Flow-like movements in granular materials such as pyroclastic terrains, are among the most destructive mass movements due to their velocity and absence of warning signs (Hutchinson, 2004; Cascini, 2008).The above mentioned studies focused on a number of sites; namely: Cervinara (Damiano & Mercogliano 2013; Rianna et al. 2016), Nocera Inferiore (Reder et al. 2016; Rianna et al. 2017a, 2017b) and Ravello (Ciervo et al. 2016). Several aspects differentiate the case studies and, consequently, the respective investigations. Depth, stratigraphy and grain size of pyroclastic covers are fundamentally regulated by slope, distance to volcanic centers (Campi Flegrei and Somma-Vesuvio), as well as wind direction and magnitude during the eruptions; therefore, the critical rainfall pattern inducing slope failure varies according to these differences (e.g. intensity, length of antecedent precipitation time window). Indeed, at some locations (Cervinara and Nocera Inferiore), the triggering event is recognized to be characterized by daily duration (also acting jointly with wet antecedent conditions), while in other locations (Ravello) events in shallower covers or covers formed by coarser material require heavy rainfall lasting several hours.  Consequently, two different approaches are followed for (the scope of such a data elaboration) based on the considered duration. The former relative to daily durations is modifying daily observations according to projected anomalies (Damiano & Mercogliano 2013) or simulated data through statistical bias correction approaches (adopted for the Cervinara and Nocera Inferiore test cases). In the latter case, a stochastic approach is coupled with bias-corrected climate data to provide assessments at hourly scale (adopted for the Ravello test case). Some studies (Reder et al. 2016; Ciervo et al. 2016; Rianna et al. 2017b), make use of expeditious statistical approaches referring to rainfall thresholds to assess slope stability conditions, while other studies employ physically based approaches (Damiano & Mercogliano 2013; Rianna et al. 2017a).

## 1.2 Object of the study

This study focuses on the quantitative estimation of the temporal evolution of hazard for rainfall-triggered flow-like movements affecting a stretch of a national motorway in the municipality of Nocera Inferiore. Flow-like landslide runout is

probabilistically investigated through a frequentist estimate of "reach probability" (Rouiller et al., 1998; Copons and Vilaplana, 2008) performed in a GIS environment, thus allowing the seamless mapping of landslide hazard under current and future climate change scenarios

The study presents significant elements of novelty. For instance, through a Bayesian approach, it characterizes precipitation values cumulated on two time windows as proxies for the triggering of flow-like movements in pyroclastic covers in the Monti Lattari mountain chain. The resulting quantitative model returns temporal variations in triggering probability, thus accounting for the effect of climate change on rainfall trends. Uncertainties in variations in rainfall patterns are taken into account by means of the EURO-CORDEX ensemble. Projections provided by climate simulations are bias-adjusted, allowing the comparison with available physically-based rainfall thresholds while adding further assumptions and uncertainties in simulation chains.. The analysis also relies on rainfall data from the rain gauges located in Gragnano and Castellammare di Stabia. The location of the three towns in Italy and in the Campania region is illustrated in Figure 1.

## 2 Description and modelling of the study area

### 2.1 Geographic and geomorphological description

Most of the territory of the Nocera Inferiore municipality belongs geomorphologically to the Sarno river valley. The most urbanized area of the town is located at the toe of the northern slopes of the Mount Albino relief, pertaining to the Monti Lattari chain (Figure 2, sector A); other more sparsely populated areas are located at the foot of the Torricchio hills (Figure 2, sector B). These reliefs are constituted by carbonate rocks covered by air-fall pyroclastic deposits originated from volcanic eruptions (Somma-Vesuvio complex) during the last 10,000 years (Pagano et al., 2010). Such covers in loose pyroclastic soils have been historically affected by multiple types of rainfall-induced flow movements,including Gragnano (in 1997), Sarno & Quindici (in 1998), Nocera (in 2005) and Ischia (in 2006). The complete list of events affecting the area considered in the study in the period 1960-2015 is given in Table 2.  Movement types include: (a) "hyper-concentrated flows" (flows in transition from mass transport to mass movement) which are generally triggered by washing away and/or progressive erosive processes along rills and inter-rill areas; (b) "channelized debris flows" (channelized flow-like mass movement) which are generated by slope failure in "zero order basin" ZOB areas (Dietrich et al. 1986; Cascini et al. 2008) and funnel into stream channels (the propagation process is laterally confined by the stream channel); and (c) "un-channelized debris flows" (un-channelized flow-like mass  movement) (Costa 1984, Hutchinson et al. 2004), which are locally triggered on open-slope areas and propagate as debris avalanches (the propagation process is not laterally confined). The latter type characterized the most recent event which affected the town in March 2005, causing three fatalities and extensive damage to buildings and infrastructures (Pagano et al. 2010; Rianna et al. 2014). This study focuses specifically on a section of the Autostrada A3 "Salerno-Napoli" motorway, which runs across the toe of the Monte Albino relief as shown in Figure 3.

# 3 Hazard: glossary and model

In the disaster risk management discipline, the term "hazard" has been linked to multiple definitions and operational guidelines, depending on the scope, technical approach employed and available data and models. Fell et al. (2008) distinguish "susceptibility zoning" from "hazard zoning" on the basis that the former involves the spatial distribution and rating of the terrain units according to their propensity to produce landslides", while the latter should include, wherever possible, the quantitative estimation of the frequency of landsliding. In situations where this estimation is difficult to pursue, some approximate guidance on frequency should be provided. Corominas et al. (2005) defined hazard as "a condition that expresses the probability of a particular threat occurring within a defined time period and area", with no explicit reference to frequency. In this study, calculated hazard levels refer to a frequency of the occurrence at least one landslide event within the study area in 30-year periods. In the more specific context of the case study presented herein, the geologic hazard mapping webpage of ISPRA, the Italian governmental Institute for environmental protection and research, states that landslide prediction models "usually address where an event is expected to occur and with which probability, without explicitly estimating return periods and intensities".

Operationally, the study is conducted by coupling mathematical software with GIS to obtain spatially referenced estimates of hazard and its mapping. A digital terrain model (DTM) of the study area, having a resolution of 15x15 m, was built for the purpose of GIS-based modelling of runout. The original resolution adopted in the Regione Campania ORCA project (2004) was 5x5 m. A variety of DTM resolutions were tested for the case study. The adopted resolution proved to be sufficient to adequately represent the surface morphology and runout as detailed in Section 6. Hazard is estimated quantitatively for each cell of the GIS-generated grid through the following model:

$$H = P_L \cdot P_R \tag{1}$$

in which $P_L$ is the probability of event occurrence, and $P_R$ is the reach probability for the cell. Occurrence probability defines the likelihood of the occurrence of at least one event in the study area as a consequence of the attainment of given thresholds of cumulative rainfall and of the likelihood of triggering given the occurrence of such thresholds. Reach probability describes the probability that a given cell will be reached by a moving soil mass, assuming that flow-like landslides have been triggered in one or more potential source areas. Occurrence probability and reach probability are distinct parameters which depend from different factors and which are computed separately.

Occurrence probability is partly related to the likelihood of triggering given the attainment of specific rainfall thresholds, which is assumed to be an inherent, time-invariant attribute of the area, and partly related to climate change through the time-dependent probability of exceedance of such rainfall thresholds as described in Section 5. Reach probability is not related to climate change, as it parameterizes the probability of spatial occupation during runout, assuming that triggering has occurred. Reach probability depends solely on terrain factors, and is thus amenable to the concept of susceptibility as defined by Fell et al. (2008) among others. Occurrence and triggering probabilities are related to rainfall parameters and, thus, are assumed to be spatially invariant and uniform for the entire area (but time-dependent), while reach probability depends on

geomorphological factors, and is thus cell-specific and spatially variable (but time-invariant) within the area. These aspects are detailed further in the paper. The study is conducted according to the operational flowchart shown in Figure 4. The modular approach initially involves the disjoint estimation of occurrence probability (including its temporal variation) as described in Section 5, and of reach probability, as detailed in Section 6. Subsequently, hazard is calculated in Section 7 using the model described above.

## 4 Source datasets

### 4.1 Observed precipitation data

Observed datasets are used to identify time windows used as proxies for flow-like landslide triggering, to implement the Bayesian approach described in Section 5.2. Subsequently, data from the Nocera Inferiore station are used for the bias adjustment of climate projections in estimating occurrence probability (Section 5.3). Although the study is focuses on Nocera Inferiore flow-like movements, data from the neighbouring towns of Gragnano and Castellammare di Stabia are considered in order to increase the size of the event database, thus increasing the statistical significance of the approach. At both sites, events affecting pyroclastic covers were observed to be very similar to those of the Nocera Inferiore slopes (De Vita & Piscopo 2002) as described in Section 4.2.

The dataset related to daily precipitation spans across the time window from January 01, 1960 to December 31, 2015. Unfortunately, no weather stations were in operation throughout the entire period for any of the three towns. Consequently, the dataset was reconstructed by merging data provided by different weather stations. Prior to 1999, the network of monitoring stations was managed by Servizio Idrografico e Mareografico Nazionale (SIMN, Hydrographic and Tidal National Service) network at national level. In that period, the selected reference weather station is that located within the town and identified with the town's name as can be found in the SIMN yearbooks. Subsequently, the management was delegated to regional level, with the Regional Civil Protection managing the dataset for the Campania region. Since 1999, the reference weather stations are selected among those adopted for the towns in Regional Early Warning Systems against geological and hydrological hazards (Sistema di Allertamento Regionale per il rischio idrogeologico e idraulico ai fini di protezione civile, 2005). Checks for the homogeneity of time series and for the unwarranted presence of breakpoints between the two periods were carried out for this study through the Pettitt (1979) and CUSUM (CUmulative SUM) (Smadi & Zghoul 2006) tests. Source weather stations, location, installation time and main (i.e., at least four months in a year) out-of-use periods are reported in Table 1.

### 4.2 Flow-like movements inventory

The inventory was compiled using three main references: Vallario (2000), De Vita & Piscopo (2002) and, for the more recent events, the "Event Reports" drafted by the Regional Civil Protection. The multiple sources used for reconstructing the inventory provide quite different details. De Vita and Piscopo (2002), for example, report the cumulative rainfall values

inducing the events on time spans up to 60 days for events in the same geomorphological context. Vallario (2000) provides brief descriptions about the events (also for the other natural hazards affecting the Region) including the number of fatalities and injured. "Event Reports", drafted by the Regional Civil Protection, contain exhaustive descriptions about the weather patterns inducing the triggering event and the main consequences for the affected communities. It is worth recalling that only events affecting pyroclastic covers have been considered and included in the dataset. Sixteen events were observed in the period 1960-2015 as detailed in Table 2.

## 4.3 Climate projections

The generation of climate projections was conducted for Nocera Inferiore as a preliminary step to the quantitative characterization of the temporal evolution of occurrence probability, since the latter depends partly on the frequency with which specific rainfall thresholds are attained. The adopted simulation chain includes several elements. Firstly, scenarios about future variations in the concentrations of atmospheric gases inducing climate alterations are assessed through socio-economic approaches including demographic trends and land use changes. IPCC (Intergovernmental Panel on Climate Change) defined Reference Concentration Pathways (RCP) in terms of increases in radiative forcing in the year 2100 (compared to preindustrial era) of about 2.6, 4.5, 6.0 and 8.5 W/m$^2$. Such scenarios force Global Climate Models (GCM). These are recognized to reliably represent the main features of the global atmospheric circulation but fail to reproduce weather conditions at temporal and spatial scales of relevance for assessing impacts at regional/local scale. In order to bridge such gap, GCMs are usually downscaled through Regional Climate Models (RCMs). These are climate models nested on GCMs, from which they retrieve initial and boundary conditions, but which work at higher resolution (including a non-hydrostatic formulation) on a limited area. The dynamic downscaling from GCMs to RCMs allows a better representation of surface features (orography, land cover, etc.) and of associated atmospheric dynamics (e.g., convective processes). Nevertheless, persisting biases can hinder the quantitative assessment of local impacts.

In order to cope with such shortcomings, a number of strategies can be adopted. For instance, to characterize uncertainty associated to future projections, climate multi-models ensemble can be utilized where different combinations of GCM and RCM run on fixed grid and domain. Furthermore, statistical approaches (e.g., Maraun 2013; Villani et al. 2015; Lafon et al. 2013) can be pursued to reduce biases assumed as systematic in simulations. More specifically, quantile mapping approaches have been applied with satisfactory results in recent years for impact studies. In these applications, the correction is performed as to ensure that "a quantile of the present-day simulated distribution is replaced by the same quantile of the present-day observed distribution" (Maraun 2013). However, limitations and assumptions associated to these approaches should be clear to practitioners (Ehret 2012; Maraun & Widmann 2015).

In the present study, climate simulations included in EURO-CORDEX multi-model ensemble at 0.11' (approximately 12 km) are considered under the RCP4.5 and RCP8.5 scenarios as described in Table 3. Climate simulations are bias-adjusted through an empirical quantile mapping approach (Gudmundson et al. 2012) using data from Nocera Inferiore weather stations from the period 1981-2010.

In Figure 5, the variations expected in monthly cumulative values (5a) and maximum daily precipitations (5b) are displayed assuming 1981-2010 as reference period and splitting the period 2010-2100 in three 30-year periods. More specifically, the upper part of Figure 5a shows the expected variations in monthly cumulative variations for RCP 4.5 (continuous line) and RCP8.5 (hatched line) as returned by bias-corrected projections in the short-term (green; 2011-2040 vs 1981-2010), medium-term (blue; 2041-2070 vs 1981-2010) and long-term (red; 2071-2100 vs 1981-2010). The bottom part of Figure 5a shows the observed annual cycle of monthly cumulative precipitations (in mm). Figure 5b shows the mean values of maximum daily precipitations in the reference observed period (1982-2009) and projected on short-term (green: 2011-2040 vs 1981-2010), medium-term (blue: 2041-2070 vs 1981-2010) and long-term (red: 2071-2100 vs 1981-2010). Filled and dashed bars correspond to results for RCP4.5 and RCP8.5, respectively.

The ensemble mean values from EURO-CORDEX optimally overlaps the actual values (data not displayed) for the same time span. Concerning future time periods, reductions up to 45% (under RCP8.5) are expected in the summer season. In this perspective, the decreases are mainly regulated by the severity of concentration scenarios. Values generally lower than the current ones are also estimated in spring (approximately -10%) and in the first part of autumn (approximately -5%). These predictions are characterized by a fluctuating signal. An increase is expected in the remaining seasons, with few exceptions (i.e., short term 2011-2040 under RCP4.5). Higher increases could exceed 20% in November and 15% in January. These evolutions could primarily induce variations in the timing of flow-like movements affecting pyroclastic covers in the area. Such events tend to occur especially in the second part of winter (or first part of spring) following the increase in antecedent precipitations. On the contrary, the likelihood of occurrence reduces during autumn and in the first part of winter. It is also worth noting that the expected increase in temperature (not taken into account in this approach) could lead to a higher atmospheric evaporative demand and, thus, to lower values of soil water content within the pyroclastic covers. Regarding precipitation triggering events, the variations in maximum daily precipitation are displayed in Figure 4b. Under both scenarios, increases with respect the reference value (about 90 mm/day) ranging from 5 and 15% for "mid-way" scenario and as high as 20% are expected under RCP8.5 for the intermediate time horizon.

# 5 Occurrence probability

## 5.1 Calculation method

Flow-like landslide occurrence probability was estimated quantitatively as a function of two cumulative rainfall thresholds; namely, the 1-day rainfall $\beta_{01}$ and the 59-day rainfall $\beta_{59}$. Several studies have stressed the prominent role of antecedent precipitations for occurrence of movements in pyroclastic covers: De Vita and Piscopo (2002) used 59-day rainfall for the same geomorphological context; Napolitano et al., (2016) defined different Intensity-Duration (I-D) rainfall thresholds for dry and wet seasons for the Sarno area. Comegna et al. (2017) assessed through a statistical framework that effective precipitation period for the Monti Lattari area could be 3 months long. Fiorillo & Wilson (2004) suggested a simplified approach to evaluate the attainment of soil moisture states which could act as triggering factors. Pagano et al. (2010), interpreting the 2005 events

in Nocera Inferiore, suggested that antecedent precipitations, should be considered at least 4-months long for those events. Reder et al. (2018) stressed the role of soil-atmosphere water exchanges during the entire hydrological year, accounting also for the effect of evaporation losses. They also stated that the effective length of effective antecedent precipitation window is highly dependent from local conditions: cover depth, pumice lenses, bottom hydraulic conditions.

In this study, cumulative rainfall parameters were calculated using a moving window procedure associated with each day from January 01, 1960 to December 31, 2015 from the observed precipitation data described in Section 4.1. The number of events observed for each day at the Nocera Inferiore, Gragnano and Castellammare di Stabia as reported in the inventory was associated with the rainfall data. Figure 6 plots the pairs of $\beta_{01}$ and $\beta_{59}$ recorded daily in the period 1960-2015, along with the indication of occurrence (by site) or non-occurrence of flow-like landslide events.

The probability of event occurrence is given by

$$P_L = \sum_{i=1}^{N_{\beta01}} \sum_{j=1}^{N_{\beta59}} \left[ P_T^{(ij)} \cdot P\left(\beta_{01}^{(i)}, \beta_{59}^{(j)}\right) \right] \tag{2}$$

in which

$\beta_{01}^{(i)}$            $i$-th value of cumulative rainfall $\beta_{01}$ ($i$=1,…, $N_{\beta01}$)

$\beta_{59}^{(j)}$            $j$-th value of cumulative rainfall $\beta_{59}$ ($j$=1,…,$N_{\beta59}$)

$P_T^{(ij)} = P\left(T|\beta_{01}^{(i)}, \beta_{59}^{(j)}\right)$    conditional probability of triggering of a flow-like landslide given the simultaneous occurrence of $\beta_{01}^{(i)}$ and $\beta_{59}^{(j)}$

The joint probability $P\left(\beta_{01}^{(i)}, \beta_{59}^{(j)}\right)$ of simultaneous occurrence of $\beta_{01}^{(i)}$ and $\beta_{59}^{(j)}$ is obtained as the frequentist ratio of the number of days in which the simultaneous occurrence of $\beta_{01}^{(i)}$ and $\beta_{59}^{(j)}$ was recorded to the total number of days for which observations at the rain gauges are available. While $P\left(\beta_{01}^{(i)}, \beta_{59}^{(j)}\right)$ is assumed to be temporally variable due to the climate

change-induced variations in rainfall patterns over time, triggering probability is assumed to be an inherent, temporally invariant characteristic of the study area, as it parameterizes in terms of probability the susceptibility of triggering of flow movements in the area in response to the attainment of specific rainfall thresholds. It accounts implicitly and empirically for all physical factors affecting triggering mechanisms. Triggering probability is calculated as described in the following.

**5.2 Triggering probability calculation method**

The conditional probability $P_T^{(ij)}$ of triggering of a flow-like landslide given the simultaneous occurrence of $R_{01}^{(i)}$ and $R_{59}^{(j)}$ is estimated using a Bayesian approach as suggested by Berti et al. (2012). The procedure refers to Bayes' theorem, formulated as follows:

$$P_T^{(ij)} = P\left(T|\beta_{01}^{(i)}, \beta_{59}^{(j)}\right) = \frac{P\left(\beta_{01}^{(i)}, \beta_{59}^{(j)}|T\right) \cdot P(T)}{P\left(\beta_{01}^{(i)}, \beta_{59}^{(j)}\right)} \qquad (3)$$

in which, in Bayesian glossary, $P\left(\beta_{01}^{(i)}, \beta_{59}^{(j)}|T\right)$ is the likelihood, i.e., the conditional joint probability of simultaneous occurrence of $\beta_{01}^{(i)}$ and $\beta_{59}^{(j)}$ if an event is triggered in the reference area; and $P(T)$ is the prior probability, i.e., the probability of triggering in the reference area, regardless of the magnitude of $\beta_{01}$ and $\beta_{59}$.

Let

$N_\beta$      total number of rainfall events recorded during a given reference time period

$N_L$      total number of flow-like landslides occurred during the given reference time period

$N_{\beta_{01}^{(i)}}$      number of rainfall events of a given magnitude of $\beta_{01}$ recorded during the given time reference

$N_{\beta_{59}^{(j)}}$      number of rainfall events of a given magnitude of $\beta_{59}$ recorded during the given time reference

5    The likelihood can be calculated as the product of the marginal conditional probabilities of attainment of $\beta_{01}^{(i)}$ and $\beta_{59}^{(j)}$ given the occurrence:

$$P\left(\beta_{01}^{(i)}, \beta_{59}^{(j)}|T\right) = P\left(\beta_{01}^{(i)}|T\right) \cdot P\left(\beta_{59}^{(j)}|T\right) \qquad (4)$$

The above Bayesian probabilities can be computed in terms of relative frequencies as follows:

$$P(T) = \frac{N_L}{N_\beta} \qquad (5)$$

$$P\left(\beta_{01}^{(i)}|T\right) = \frac{N_{\beta_{01}^{(i)}|T}}{N_L} \qquad (6)$$

$$P\left(\beta_{59}^{(j)}|T\right) = \frac{N_{\beta_{59}^{(j)}|T}}{N_L} \qquad (7)$$

in which

$N_{\beta_{01}^{(i)}|T}$      number of rainfall events of magnitude at least $\beta_{01}^{(i)}$ recorded during the given time reference and which resulted in the triggering of flow-like landslides

$N_{\beta_{59}^{(j)}|T}$      number of rainfall events of magnitude at least $\beta_{59}^{(j)}$ recorded during the given time reference and which resulted in the triggering of flow-like landslides

Figure 7 plots triggering probability $P_T$ as a function of 1-day and 59-days cumulative rainfall, as estimated through the
10   Bayesian approach. Possible future variations in land use/land cover features are assumed not to significantly affect proxy values. This is a simplistic hypothesis, as local conditions could substantially modify the susceptibility of the areas to event

occurrence (e.g., fires destroying vegetation). Should substantial variations in physical factors occur in the study area, a re-evaluation of triggering probability is warranted.

### 5.3 Occurrence probability outputs

Following the quantitative estimation of the site-specific triggering probability as described above, flow-slides occurrence probability was calculated using Eq. (2) for each of the 10 EURO-CORDEX ensemble models and for 10 sets of 30-year intervals from 1981-2010 to 2071-2100 for both the RCP4.5 and RCP 8.5 scenarios.

A quantitative statistical analysis was conducted with the aim of analysing ensemble outputs. The first module of the analysis consisted in the second-moment statistical characterization of the output samples. Such characterization involved the

calculation of mean, standard deviation and sample coefficient of variation (given by the ratio of the latter to the former) for the 10-valued sets of ensemble model outputs for each of the 10 30-year intervals. Figure 8 plots the temporal variation of $P_L$ for 10 sets of 30-year intervals from 1981-2010 to 2071-2100 and for the RCP4.5 and RCP 8.5 scenarios; more specifically: model outputs and ensemble means for RCP4.5 (8a), RCP8.5 (8b), and for both concentration scenarios (8c). Figure 8d plots the sample coefficient of variation for both scenarios.

For the RCP4.5 scenario, considering the running 30-year averages, visual inspection of Figure 8 suggested that all available projections predict a moderate increase in occurrence probability. A higher spread among the models is recognizable at the middle of the XXI century as parameterized by the peak in the sample coefficient of variation. Such increased spread is mainly due to the outputs of two models constantly representing, respectively, the upper and bottom boundaries of the ensemble throughout the entire investigated period. For the RCP8.5 scenario, one of the 10 ensemble models yields occurrence

probability values which progressively increase with respect to the other models over time. This leads to a marked increase in the scatter as parameterized by the sample coefficient of variation.

The second module of the statistical analysis consisted in the assessment of the existence and strength of a temporal statistical trend in occurrence probability values for the comprehensive set of output of the 10 models in the CORDEX ensemble for the 10 sets of 30-years periods. This analysis was conducted by means of two non-parametric statistical tests aimed at assessing

the statistical independence between occurrence probability and time (as parameterized by which 30-year interval to which a specific occurrence probability value pertains) through the calculation of rank correlation statistics and related p-values which parameterize the significance level at which the null hypothesis of statistical independence can be accepted. Spearman's test (Spearman 1904) entails the calculation of Spearman's rank correlation coefficient $\rho$ which measures rank correlation on a -1:1 scale (-1: full negative rank correlation; 0: no rank correlation; 1: full rank correlation) and of an associated p-value. The

output values of $\rho$ were 0.351 for RCP4.5 and 0.381 for RCP8.5. The associated p-values were calculated as $3.45 \cdot 10^{-4}$ for RCP4.5 and $9.22 \cdot 10^{-5}$ for RCP8.5, attesting to a very low significance level for the rejection of the null hypothesis of statistical independence between time and occurrence probability. Kendall's test (Kendall 1938) entails the calculation of the statistic $\tau$, which measures rank correlation on a -1:1 scale (-1: full negative rank correlation; 0: no rank correlation; 1: full rank

correlation) and of an associated p-value. The output values of $\tau$ were 0.245 for RCP4.5 and 0.277 for RCP8.5. The associated p-values were calculated as $5.42 \cdot 10^{-4}$ for RCP4.5 and $9.07 \cdot 10^{-5}$ for RCP8.5, again attesting to a very low significance level for the rejection of the null hypothesis. The non-parametric analysis thus assessed the existence of a strong statistical dependency of occurrence probability from time, thereby confirming the influence of climate change on flow-like movement hazard.

The third module consisted in the concise formulation of occurrence probability through the fitting of analytical models. The purpose of this model was to allow for a more concise forward estimation of triggering probability. In this study, the fitting of analytical models was conducted with the aim of relating analytically calculated values to specific levels of likelihood of exceedance of occurrence probability. This was achieved through quantile regression.

Quantile regression is a type of regression analysis often used in statistics and econometrics. Whereas the method of least

squares results in estimates that approximate the conditional mean of the response variable given certain values of the predictor variables, quantile regression aims at estimating any user-defined quantile of a response variable, in this case of triggering probability (Yu et al. 2003). Quantile regression implements a minimization algorithm and yields model parameters which define the analytical model for user-defined regression quantiles (corresponding to a likelihood of non-exceedance). The use of quantile regression enables to address explicitly different level of conservatism in the output models, with higher quantiles

corresponding to higher levels of conservatism. Quantiles of 0.50 and 0.90 were considered, corresponding to 50% and 10% likelihoods of exceedance, i.e., to scenarios of medium and high conservatism, respectively.

In applying quantile regression, a variety of analytical models were adapted to the dataset, including the linear, power, logarithmic and modified geometric models. Among these, the latter displayed the best goodness-of-fit. The modified geometric model employed in this study is given by

$$P_L = p_1 \cdot (10 \cdot t_{30})^{\frac{p_2}{t_{30}}} \tag{8}$$

in which $p_1$ and $p_2$ are the model parameters to be estimated using quantile regression and $t_{30}=1\ldots10$ is an auxiliary discrete natural variable referring to the ordinality of the 30-year averaging interval (e.g., 1981-2010 is interval "1", 2071-2100 is interval "10"). Figure 9a and Figure 9b show the quantile regression-based fits of the modified geometric model to the samples

of occurrence probability values for likelihoods of exceedance of 50% ($Q_{50}$) and 10% ($Q_{90}$) for RCP4.5 and RCP8.5, respectively.

The output model parameters for RCP4.5 were $p_1=1.38 \cdot 10^{-3}$, $p_2=-0.087$ for $Q_{50}$ and $p_1=1.71 \cdot 10^{-3}$, $p_2=-0.156$ for $Q_{90}$. For RCP8.5, $p_1=1.37 \cdot 10^{-3}$, $p_2=-0.110$ for $Q_{50}$ and $p_1=1.83 \cdot 10^{-3}$, $p_2=-0.190$ for $Q_{90}$. While the plots show a continuous fitted model for the sake of visual appreciation of the quantile regression outputs, it is to be remarked that $t_{30}$ is a discrete variable which

can only take integer values between 1 and 10. Table 4 illustrates the values of occurrence probability as calculated from the modified geometric models for $Q_{50}$ and $Q_{90}$. The ratios of occurrence probability for a given interval to that for the observed data (1981-2010) are also provided to provide a quantitative measure of the effect of climate change over time. The findings

displayed comparable increases under both RCPs with no clear increases for the more severe scenario. Such result is consistent with variations shown in Figure 4 where monthly anomalies and future expected values in maximum daily precipitations are reported. While decreases during the dry season are clearly more remarkable under RCP8.5, increases during the autumn and winter seasons do not return clear patterns regulated by scenario or time horizon. In this perspective, no significant differences between RCPs are observed.

It is worth recalling that the present approach neglects several dynamics (e.g. effects of evapotranspiration reducing soil moisture), which could have a significant role because of increased warming. For any given time interval and level of conservatism, occurrence probability is assumed to be spatially uniform within the study area, since the database which is used to develop the Bayesian method refers to the entire area itself. As detailed in a similar study by Berti et al. (2012), the quantitative output of empirical methods such as the one developed in the paper implicitly accounts for the spatial variability (if any) of rainfall characteristics within the area. In this study, three distinct reference weather stations were used for the three towns. The analysis of Nocera relies only on the local weather station, whose data were also used for bias correction purposes. Given the limited geographical extension of the area, the component of epistemic uncertainty due to spatial variability is not expected to be significant.

## 6 Reach probability

Investigation of the spatial variability of flow-like landslide hazard entails the modelling of its downslope propagation (runout). Reach probability is the probability (from 0: certainty of no reach; 1: certainty of reach) of each point in the spatial domain being affected by the event during the runout process. Several morphological, empirical and physically-based approaches are available for quantitative runout analysis (Hürlimann et al. 2008). Each of these may present advantages or weaknesses in relation to site- and/or phenomenon-specific attributes, data availability and scale of the analysis. Consistently with the methods previously used to define triggering-rainfall scenarios, the approach used to define downslope runout scenarios is based on an algorithm involving stochastic modelling.

### 6.1 Reach probability calculation method

Reach probability was computed spatially using Flow-R, a DTM-based distributed empirical model developed in the Matlab® environment (Horton et al. 2013). Due to the large geographical scale of the area and to the deep complexity of the analyzed phenomena, a not highly parameter-dependent approach was deliberately adopted. A variety of DTM resolutions were tested for the case study and a 15x15 m resolution was chosen. Comparing the DTM with the real current morphological shape of the areas both numerically and by expert judgment, the adopted resolution is deemed to represent with a good accuracy the channelized shape and the fan areas, confirming the Horton et al. (2013) observations. The flow-slide spreading is controlled by a flow direction algorithm that reproduces flow paths (Holmgren 1994) and by a persistence function to consider inertia and abruptness in change of the flow direction (Gamma 2000). The flow direction algorithm proposed by Holmgren (1994),

in the setting used in this study (x=1, see Eq. (3) in Horton et al. 2013) is similar to the multiple D8 of Quinn et al. (1991, 1995). The multiple D8 distributes the flow to all neighbouring downslope cells weighted according to slope. The algorithm tends to produce more realistic looking spatial patterns than the simple D8 algorithm by avoiding concentration to distinct lines (Seibert & McGlynn 2007). The maximum possible runout distances are computed by means a *simplified friction-limited*
*model* based on a unitary energy balance (Horton et al. 2013).

One-run propagation simulation provides possible flow-paths generated from previously identified triggering/source areas. In this work, source areas were identified by means of the official geo-morphological map of the "Campania Centrale" River Basin Authority (PSAI 2015). The set of source areas coincides with the union of the "zero order basin" (ZOB) and current "niche/failure" areas as shown in Figure 10. This hypothesis is in accordance with the requirement of consistency with accounts
of historical events and with the aim to consider the most pessimistic possible triggering scenarios (i.e., those with maximum mass potential energy).

The reach probability for any given cell $P_R$ is calculated by the following equation:

$$P_R = \frac{p_u^{fd} p_u^p}{\sum_{v=1}^{8} p_v^{fd} p_v^p} p_0 \tag{9}$$

where $u$ and $v$ are the flow directions; $p_u$ is the probability value in the $u$-th direction; $p_u^{fd}$ is the flow proportion according to the flow direction algorithm; $p_u^p$ is the flow proportion according to the persistence function; and $p_0$ is the probability
determined in the previous cell along the generic computed path. The values are subsequently normalized. Runout routing is stopped when: (1) the angle of the line connecting the source area to the most distant point reached by the flow-slide along the generic computed path is smaller than a predefined *angle of reach* (Corominas 1996); and (2) the velocity exceeds a user-fixed maximum value or is below the value corresponding to the maximum energy lost due to friction along the path. The values which do not fit the above-mentioned requirements are redistributed among the active cells to ensure conservation of the total
probability value.

## 6.2 Reach probability outputs

The propagation routine was applied to the DTM described in Section 3. An *angle of reach* of 4° was calibrated based on the geo-morphological information (i.e., the extension of the slope fan deposition) and the official hazard maps of the Landslide Risk Management Plan of the River Basin Authority (PSAI, 2015) shown in Figure 10, considering a "paroxysmal" event.
Consistently with the mean values reported by the scientific literature (Faella & Nigro 2001; Revellino et al. 2004) for the same phenomena and in the same region, the maximum runout velocity was set at 10 m/s. Figure 11 illustrates the spatial distribution of reach probability at hillslope scale. Source areas are also indicated. The runout characteristics of the event types considered (types "b" and "c", see Section 2.1) can be significantly different. Nevertheless, the same set of parameters (reach angle, velocity) satisfies both event conditions adequately. It is remarked that one un-channelized event (March 2005) was
considered in this study.

In this area, the highway runs mostly on a soil embankment. The road level is generally elevated with respect to the paths of the downslope flows. The propagation impacts the embankment and stops in front of - or laterally continues according to - the topographic information and the model setting. Differently, in some points, the highway runs approximately at the same level of the fans, thereby allowing the propagating flow to invade the road. In both cases, damage or disruptions may be caused to the infrastructure. In order to overcome this distinction and to cover both scenarios, only flow propagation to the upstream boundary of the infrastructure are considered in the study. An illustrative example is shown in the magnified focus area in Figure 12. Due to the reasons mentioned above, the road surface is only partially affected by the flow-slides. This study focuses on a 400-meter stretch of the infrastructure (from point A to point B in Figure 12), the runout values to be considered in the risk assessment should be taken along the section A-B (Figure 12). The results shown in Figure 13 attest to the marked spatial variability of reach probability along the investigated section of the A3 motorway infrastructure.

## 7 Calculation of hazard

Once occurrence probability and reach probability have been estimated as illustrated in Section 5 and Section 6, respectively, it is possible to calculate hazard using Eq. (1). Hazard is temporally variable because occurrence probability displays temporal variability as a consequence of climate change as shown in Section 5.3. Reach probability is assumed to be temporally invariant as it is deterministically related to terrain morphology. This entails that the reach probability outputs obtained in Section 6.2 are valid only for the current terrain morphology. Should significant variations in terrain morphology occur, for instance, in case of the occurrence of flow-like landslides, reach probability would need to be reassessed as described in Section 6.1.

To complete the flowchart shown in Figure 4, an example calculation of hazard is provided for the section A-B. Figure 14 shows the spatially and temporally variable hazard profile for time intervals 1991-2020 and 2071-2100, for both quantiles $Q_{50}$ and $Q_{90}$ and for RCP4.5 and RCP 8.5. The occurrence probability values used to multiply the reach probability values shown in Figure 13 are taken from Table 4. Thus, the estimated variations primarily reflect changes in occurrence probability due to expected climate changes. In this regard, an increase is estimated under both concentration scenarios, Trends in increase are related to the severity of scenario (the more severe, the higher the increase) and investigated percentile (the less frequent, the higher the increase). In spatial terms, more pronounced increases in hazard are detectable in the current peak reach probability, which increases from $4 \cdot 10^{-4}$ to $5e-4 \cdot 10$.

## 8 Concluding remarks

This paper has illustrated an innovative methodology for the quantitative estimation of rainfall-induced flow-like movement hazard. An example application of the proposed method was conducted for a short section of a motorway. The quantitative approach to hazard estimation was pursued through the implementation of models which are capable of simulating both deposition and entrainment, similarly to other notable literature contributions (e.g., Deangeli 2008; Rosatti and Begnudelli

2013; Frank et al. 2015; Stancanelli et al. 2015; Cuomo et al. 2016; Gregoretti et al. 2018). Despite the limited extension of the study area, the results displayed a marked temporal and spatial variability of hazard. The temporal variability of hazard is a consequence of climate change as parameterized through quantitative projections for concentration scenarios RCP4.5 and RCP8.5. Significant temporal variability was assessed for both concentration scenarios. The considerable spatial variability

resulting from the case study stems from the spatial variability of reach probability as modelled in the runout analysis.

The calculation of occurrence probability, specifically in the triggering probability calculation phase, relies on a Bayesian approach which replicates the one provided by Berti et al. (2012). This study replicates the hypotheses and glossary introduced by these Researchers, and shares the implications, and possible limitations of such approach. For instance, the modelling hypothesis by Berti et al. (2012) is adopted, by which multiple flow-like landslides are counted as one single event. Hence, the

Bayesian method presented in the paper quantifies the probability of occurrence of a landslide scenario (defined as "at least one event in the proximity area") in a well-defined time period (30 years). This definition of scenario is semantically consistent with the one provided by Corominas et al.,  Reach probability as estimated quantitatively in the study is consistent with this definition, as it is calculated from the superposition of all possible runout paths from all events potentially occurring from all source areas. Hazard as calculated using the above hypotheses is thus a conservative, upper-bound estimate related to a specific

rainfall scenario involving specific values of 1-day and 59-day cumulative rainfall. From a semantic standpoint, hazard as defined herein is also consistent with the reference glossary provided by Corominas et al. (2005).

The quantitative estimates of hazard as obtained in this paper are pervaded by significant uncertainty. Among the main sources of uncertainty are the climate change projections, the runout model and the Bayesian model developed to quantify triggering probability. These uncertainties are epistemic in nature, as they stem from the inherent difficulty in compiling climate change

projections, the inevitable degree of approximation and imperfection in runout modelling capabilities, the limited rainfall and flow-like landslide occurrence data used to develop triggering probability curves. As such, increased modelling capability and improved databases could reduce the magnitude of uncertainty associated with hazard estimation.

The hazard outputs obtained by the method can be used directly in the quantitative estimation of risk. The latter also requires the quantitative estimation of the vulnerability of human-valued assets (i.e., vehicles, persons, etc.) and the exposure (i.e., the

number and/or degree of presence) of the assets themselves in the study area in a reference time period.

Notwithstanding the above uncertainties and limitations, the quantitative estimation and assessment of the spatial and temporal variability of hazard provide an important decision support tool in the disaster risk management cycle; specifically, in the planning and prioritization of hazard mitigation and risk mitigation measures. The availability of quantitative methods allows a more rational decision-making process in which the costs and effectiveness of risk mitigation can be compared and assessed.

Campanian pyroclastic covers are characterized by several specific features (high porosity, significant water retention capacity, intermediate saturated hydraulic conductivities) playing a relevant role for triggering (e.g. role of antecedent precipitations or persistency/magnitude of potential triggering event). Moreover, stratigraphic details as the actual grain size distribution, the presence of pumice lenses or the depth of pyroclastic deposits regulated by the distance from the eruptive centers and wind direction/magnitude during the eruptions make complex also generalisations within the same Campania Region. Nevertheless,

the framework developed for the pyroclastic covers on the North side of the Monti Lattari (where Nocera Inferiore is located) appears easily transferable to other contexts where precipitation observations and details about the timing of flow-like movements are available. Similarly, the climate simulation chain follows the state-of-the-art for analysis of impacts potentially induced by climate changes. Finally, the estimated increases in hazard result consistent with those reported in several works investigating the variation in frequency of events in coarse grained soils (Gariano & Guzzetti, 2016).

**Acknowledgments**

The research leading to these results has received funding from the European Union Seventh Framework Program (FP7/2007-2013) under grant agreement No. 606799. The support is gratefully acknowledged.

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

Table 1. Weather stations used in the compilation of datasets for Nocera Inferiore, Gragnano and Castellammare di Stabia: location, installation time and main out-of-use periods

| Town | Weather station (1960-1999) | Installation and main out-of-use periods | Weather Station (2000-2015) | Installation and main out-of-use periods |
|---|---|---|---|---|
| Nocera Inferiore | Nocera Inferiore (61 m asl) 40° 45' 0'' N 14° 38' 9'' E | Since 1899 1964,1965,1967, 1981,1982 | Tramonti (422 m asl) 40° 42' 14" N 14° 38' 49" E | Since February 2002 2000,2001 |
| Gragnano | Gragnano (173 m asl) 40° 40' 59''N 14° 31' 9'' E | Since 1921 | Gragnano_2 (195 m asl) 40° 41' 15" N 14° 31' 38" E | Since November 2001 2000,2001 |
| Castellammare di Stabia | Castellammare di Stabia (18 m asl) 40° 41' 30''N 14° 28' 17''E | Since 1929 1964,1965,1966 | Pimonte (437 m asl) 40° 40' 27" N  14° 30' 17" E | Since October 2000 2000 |

Table 2. Flow-like mass movements affecting pyroclastic covers in Nocera Inferiore, Gragnano and Castellammare di Stabia in the period 1960-2015

| Nocera Inferiore | Gragnano | Castellammare di Stabia |
| --- | --- | --- |
| 8 December 1960 | 17 February 1963 | 17 February 1963 |
| 4 November 1961 | 2 January 1971 | 17 November 1985 |
| 6 March 1972 | 21 January 1971 | 23 February 1987 |
| 10 January 1997 | 22 February 1986 | 10 November 1987 |
| 4 March 2005 | 10 January 1997 | 11 January 1997 |
|  | 4 March 2005 |  |

Table 3. Available Euro-CORDEX simulations at a 0.11º resolution (~12km) over Europe, providing institutions, GCM and RCMs

| Code | Institution | GCM | RCM |
| --- | --- | --- | --- |
| 1 | CLMcom | CNRM-CM5_r1i1p1 | CCLM4-8-17_v1 |
| 2 | CLmcom | EC-EARTH_r12i1p1 | CCLM4-8-17_v1 |
| 3 | CLMcom | MPI-ESM-LR_r1i1p1 | CCLM4-8-17_v1 |
| 4 | DMI | EC-EARTH_r3i1p1 | HIRHAM5_v1 |
| 5 | KNMI | EC-EARTH_r1i1p1 | RACMO22E_v1 |
| 6 | IPSL-INERIS | IPSL-CM5A-MR_r1i1p1 | WRF331F_v1 |
| 7 | SMHI | CNRM-CM5_r1i1p1 | RCA4_v1 |
| 8 | SMHI | EC-EARTH_r12i1p1 | RCA4_v1 |
| 9 | SMHI | MPI-ESM-LR_r1i1p1 | RCA4_v1 |
| 10 | SMHI | IPSL-CM5A-MR_r1i1p1 | RCA4_v1 |

Table 4. Temporal evolution of occurrence probability for RCP4.5 and RCP8.5 (50th and 90th quantiles)

| Interval | RCP4.5 | | | | RCP8.5 | | | |
|---|---|---|---|---|---|---|---|---|
| | $P_L(Q_{50})$ | ratio | $P_L(Q_{90})$ | ratio | $P_L(Q_{50})$ | ratio | $P_L(Q_{90})$ | ratio |
| 1981-2010 | $1.13 \cdot 10^{-3}$ | 1.00 | $1.20 \cdot 10^{-3}$ | 1.00 | $1.06 \cdot 10^{-3}$ | 1.00 | $1.18 \cdot 10^{-3}$ | 1.00 |
| 1991-2020 | $1.21 \cdot 10^{-3}$ | 1.07 | $1.36 \cdot 10^{-3}$ | 1.13 | $1.16 \cdot 10^{-3}$ | 1.09 | $1.37 \cdot 10^{-3}$ | 1.17 |
| 2001-2030 | $1.25 \cdot 10^{-3}$ | 1.11 | $1.44 \cdot 10^{-3}$ | 1.20 | $1.21 \cdot 10^{-3}$ | 1.14 | $1.47 \cdot 10^{-3}$ | 1.25 |
| 2011-2040 | $1.27 \cdot 10^{-3}$ | 1.13 | $1.49 \cdot 10^{-3}$ | 1.24 | $1.24 \cdot 10^{-3}$ | 1.16 | $1.53 \cdot 10^{-3}$ | 1.30 |
| 2021-2050 | $1.29 \cdot 10^{-3}$ | 1.14 | $1.52 \cdot 10^{-3}$ | 1.27 | $1.26 \cdot 10^{-3}$ | 1.18 | $1.57 \cdot 10^{-3}$ | 1.33 |
| 2031-2060 | $1.30 \cdot 10^{-3}$ | 1.15 | $1.54 \cdot 10^{-3}$ | 1.29 | $1.27 \cdot 10^{-3}$ | 1.20 | $1.60 \cdot 10^{-3}$ | 1.36 |
| 2041-2070 | $1.31 \cdot 10^{-3}$ | 1.16 | $1.56 \cdot 10^{-3}$ | 1.30 | $1.28 \cdot 10^{-3}$ | 1.21 | $1.63 \cdot 10^{-3}$ | 1.38 |
| 2051-2080 | $1.31 \cdot 10^{-3}$ | 1.16 | $1.57 \cdot 10^{-3}$ | 1.31 | $1.29 \cdot 10^{-3}$ | 1.21 | $1.65 \cdot 10^{-3}$ | 1.40 |
| 2061-2090 | $1.32 \cdot 10^{-3}$ | 1.17 | $1.59 \cdot 10^{-3}$ | 1.32 | $1.30 \cdot 10^{-3}$ | 1.22 | $1.66 \cdot 10^{-3}$ | 1.41 |
| 2071-2100 | $1.32 \cdot 10^{-3}$ | 1.17 | $1.60 \cdot 10^{-3}$ | 1.33 | $1.31 \cdot 10^{-3}$ | 1.23 | $1.67 \cdot 10^{-3}$ | 1.42 |

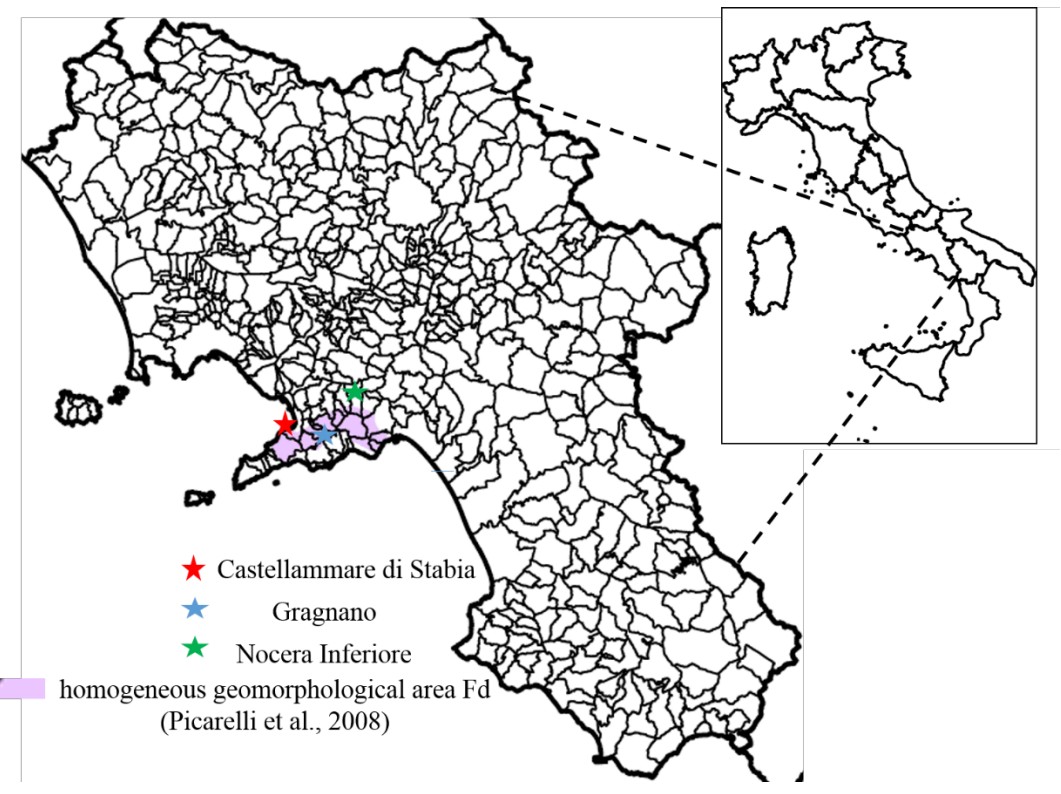

Figure 1. Identification of the three towns considered in the study

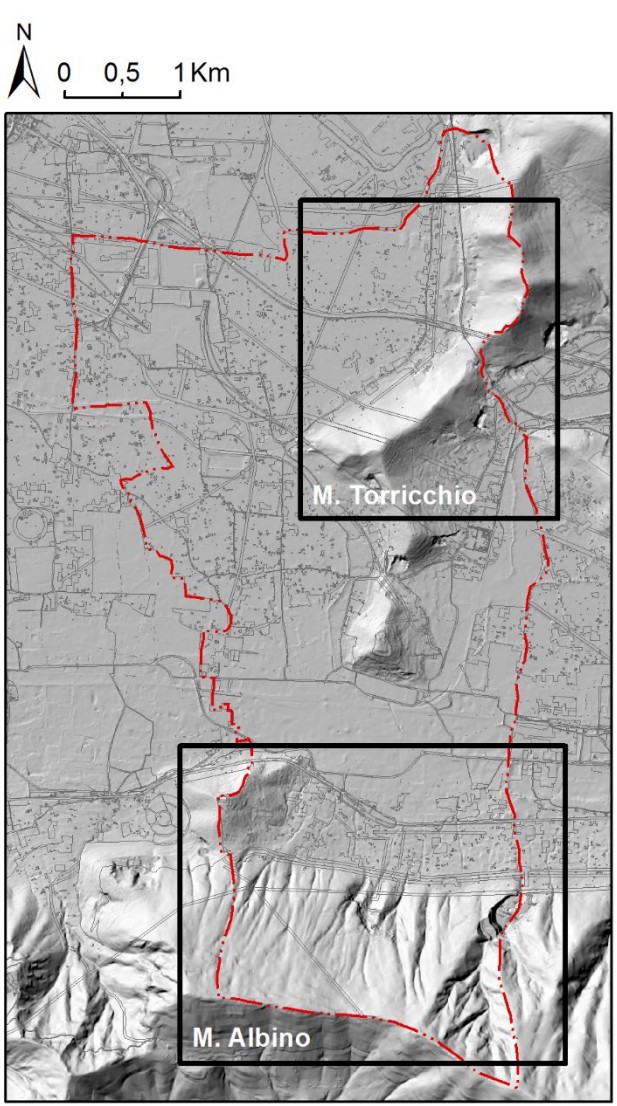

Figure 2. Geomorphologic setting and administrative boundaries of the Nocera Inferiore municipality

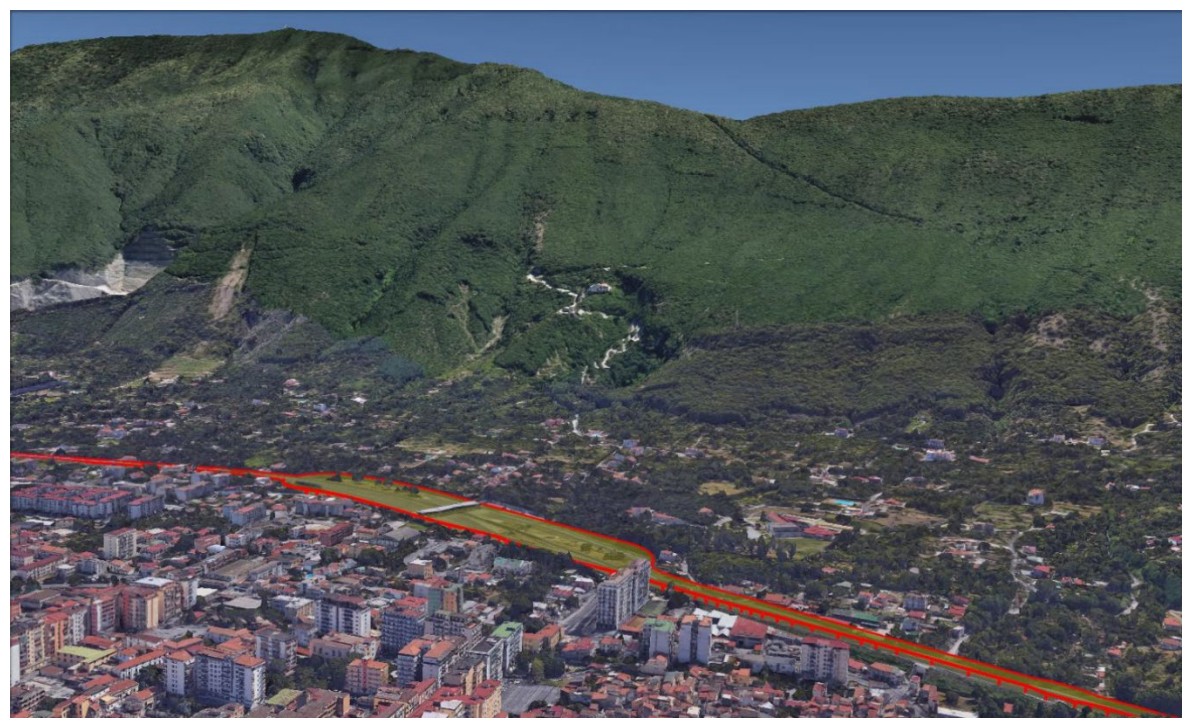

Figure 3. Infrastructure-scale view of the study area with the A3 Salerno-Reggio Calabria motorway (boundaries marked in red)

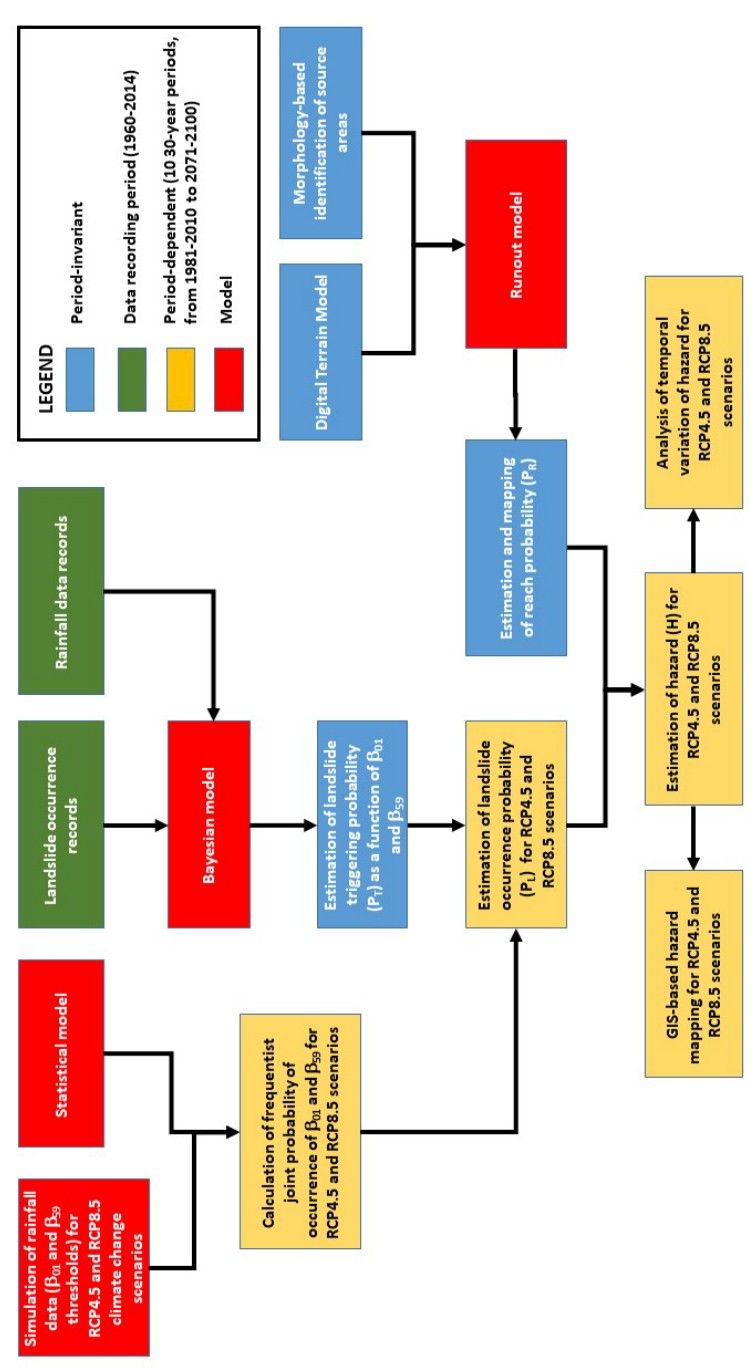

Figure 4. Operational flowchart of the study

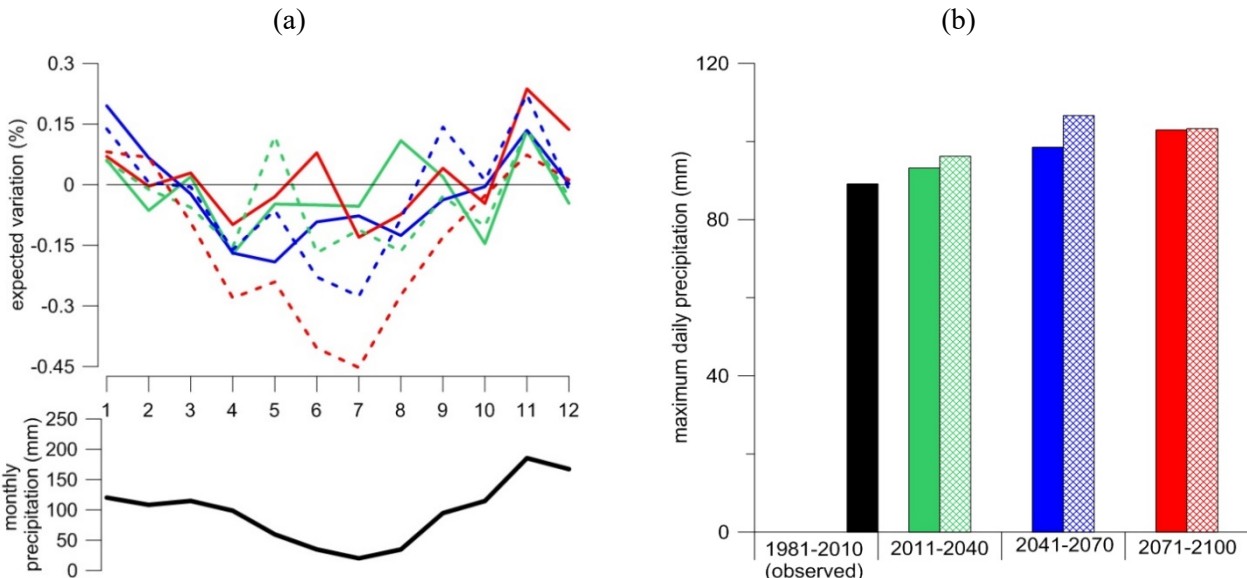

Figure 5 (a): expected variations in monthly cumulative variations; (b): mean values of maximum daily precipitations.

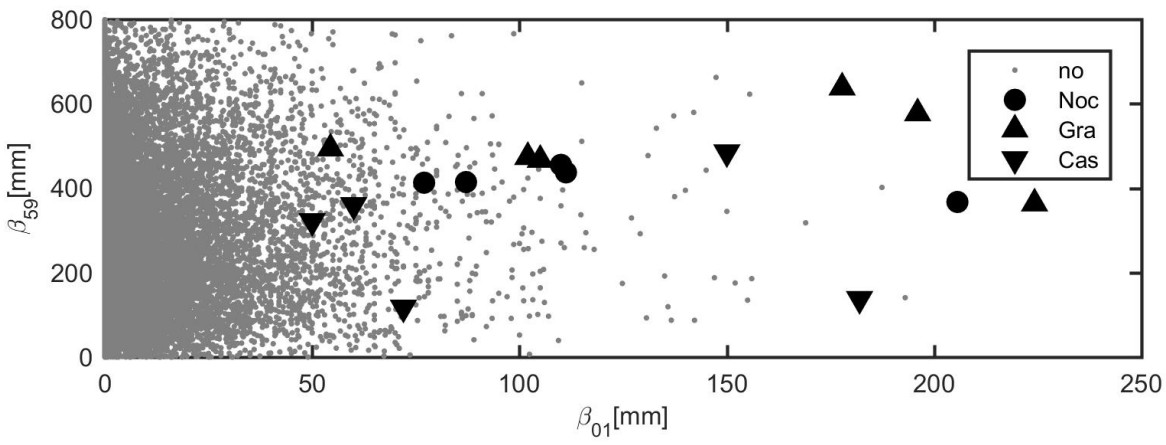

Figure 6. Pairs of $\beta_{01}$ and $\beta_{59}$ recorded daily in the period 1960-2015, with occurrence (by site) or non-occurrence of landslide events

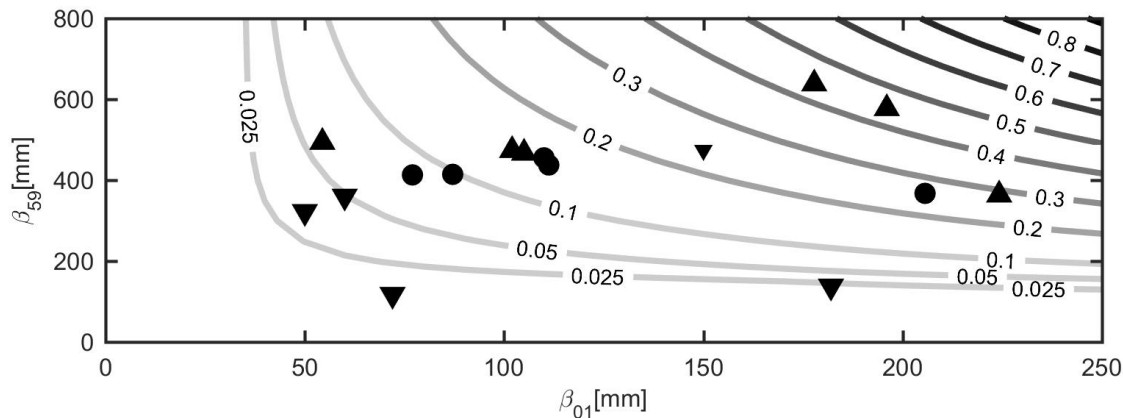

Figure 7. Landslide triggering probability $P_L$ as a function of 1-day and 59-days cumulative rainfall

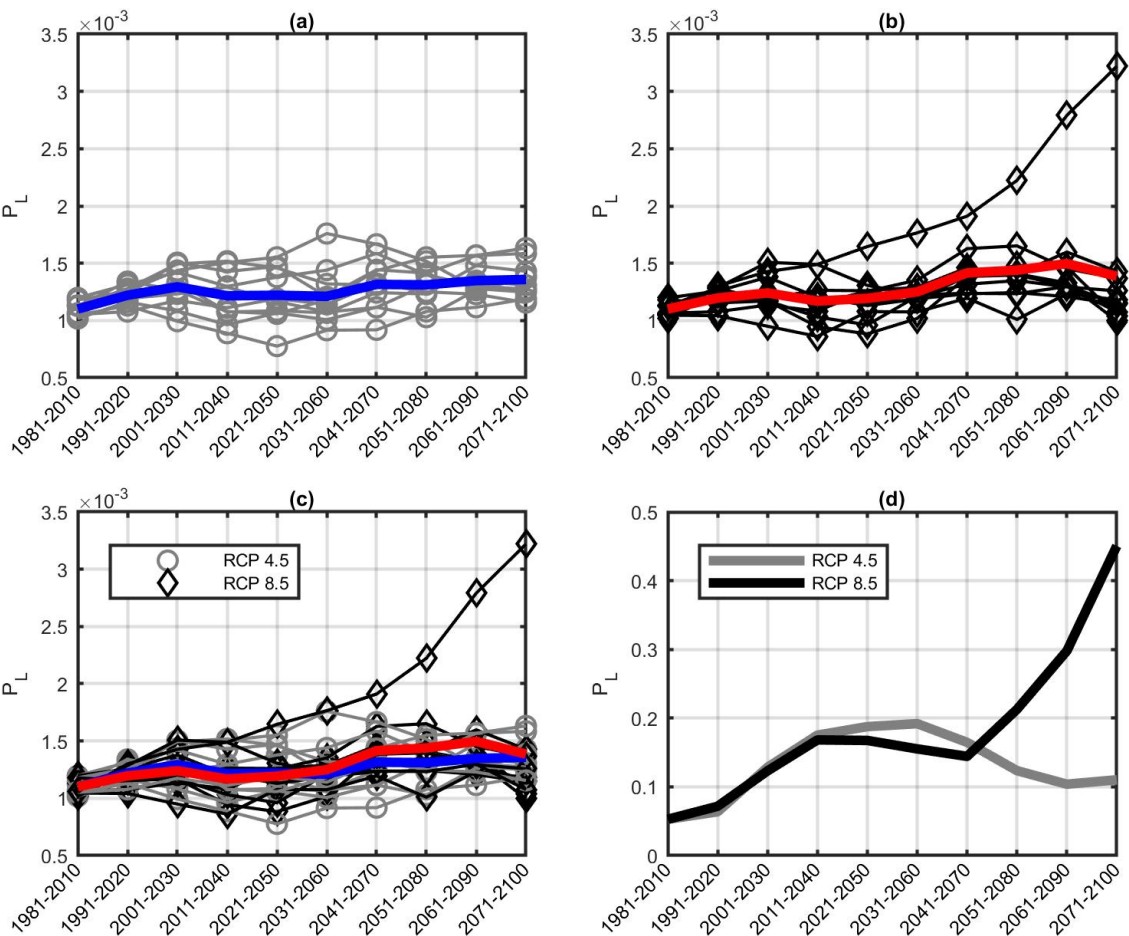

Figure 8. Outputs of second-moment statistical analysis of landslide occurrence probability $P_L$

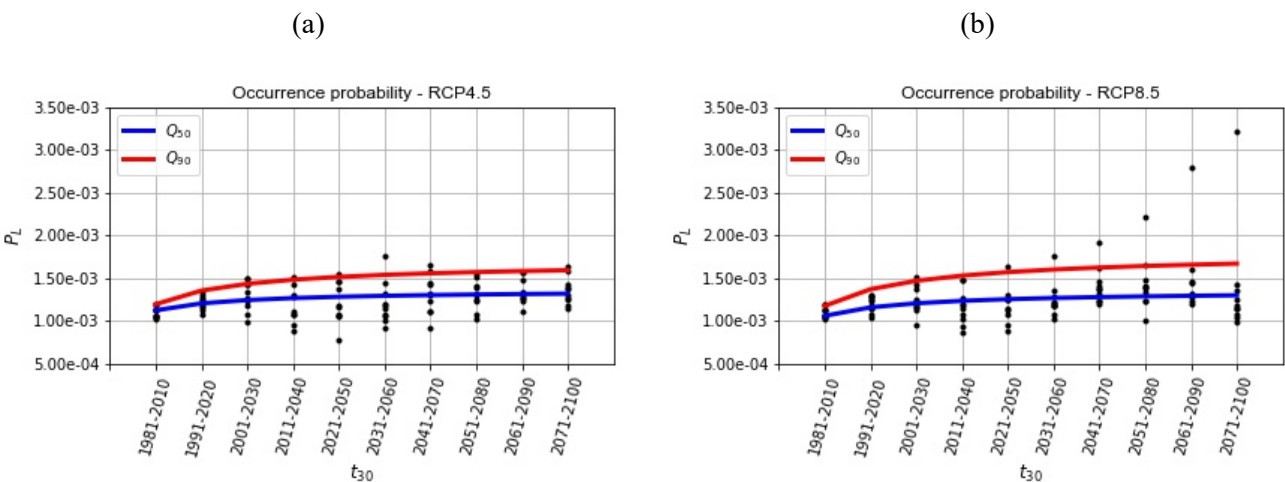

Figure 9. Fitting of modified geometrical models to landslide occurrence probability ensemble data for quantiles $Q_{50}$ and $Q_{90}$: (a) RCP4.5; and (b) RCP8.5

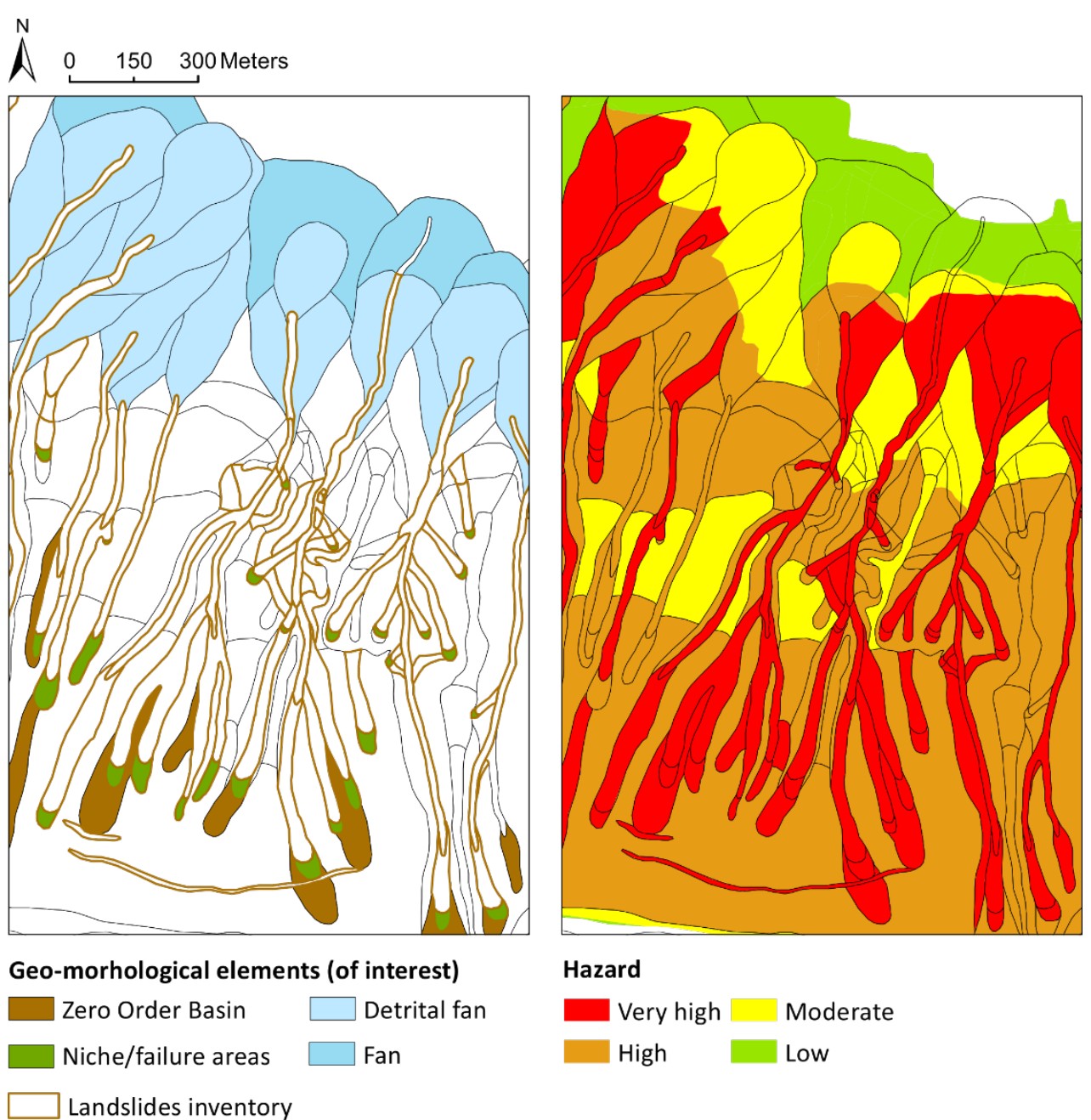

**Geo-morhological elements (of interest)**

- ⬛ Zero Order Basin
- 🟩 Niche/failure areas
- ⬜ Landslides inventory
- 🟦 Detrital fan
- 🟦 Fan

**Hazard**

- 🟥 Very high
- 🟧 High
- 🟨 Moderate
- 🟩 Low

Figure 10. Geo-morphological (left) and Hazard maps (right) of the Landslide Risk Management Plan of the River Basin Authority (PSAI, 2015

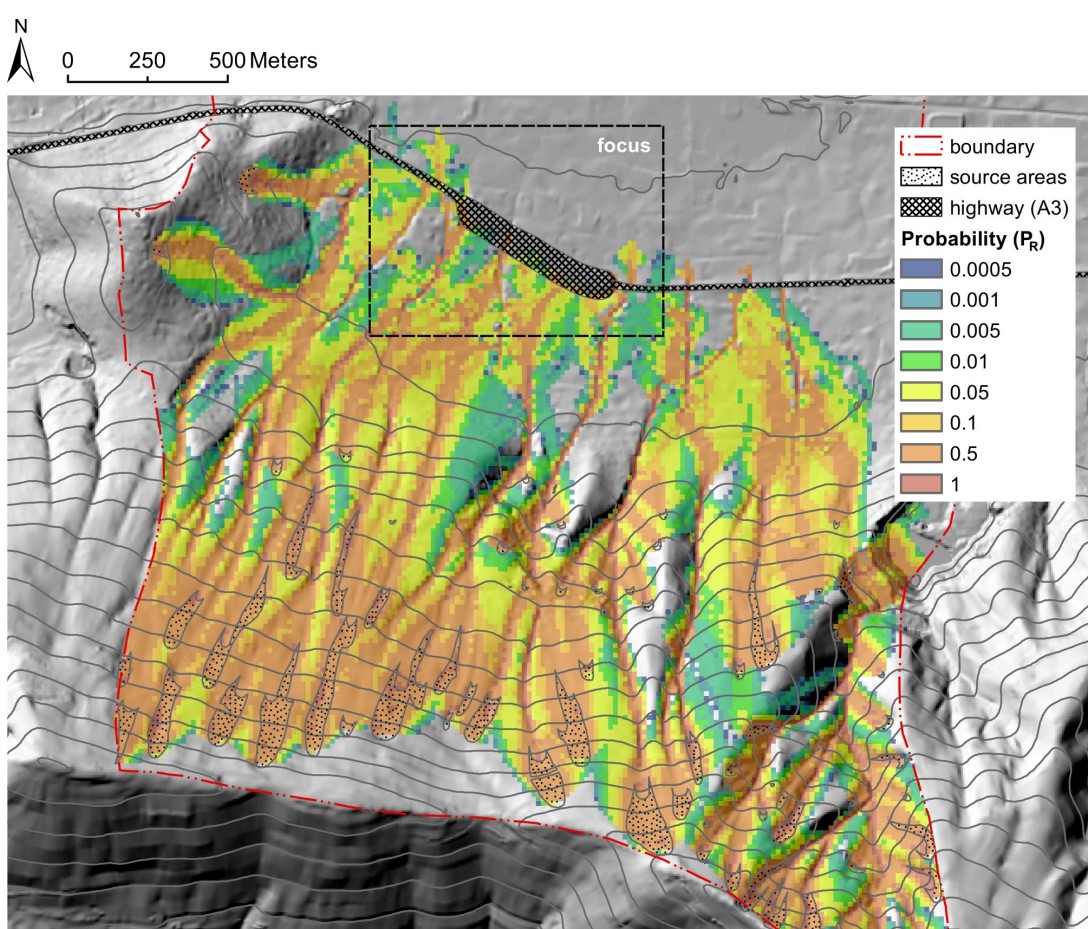

Figure 11. Spatial distribution of reach probability at hillslope scale; the area corresponds to the box named "Mt. Albino" in Figure 2

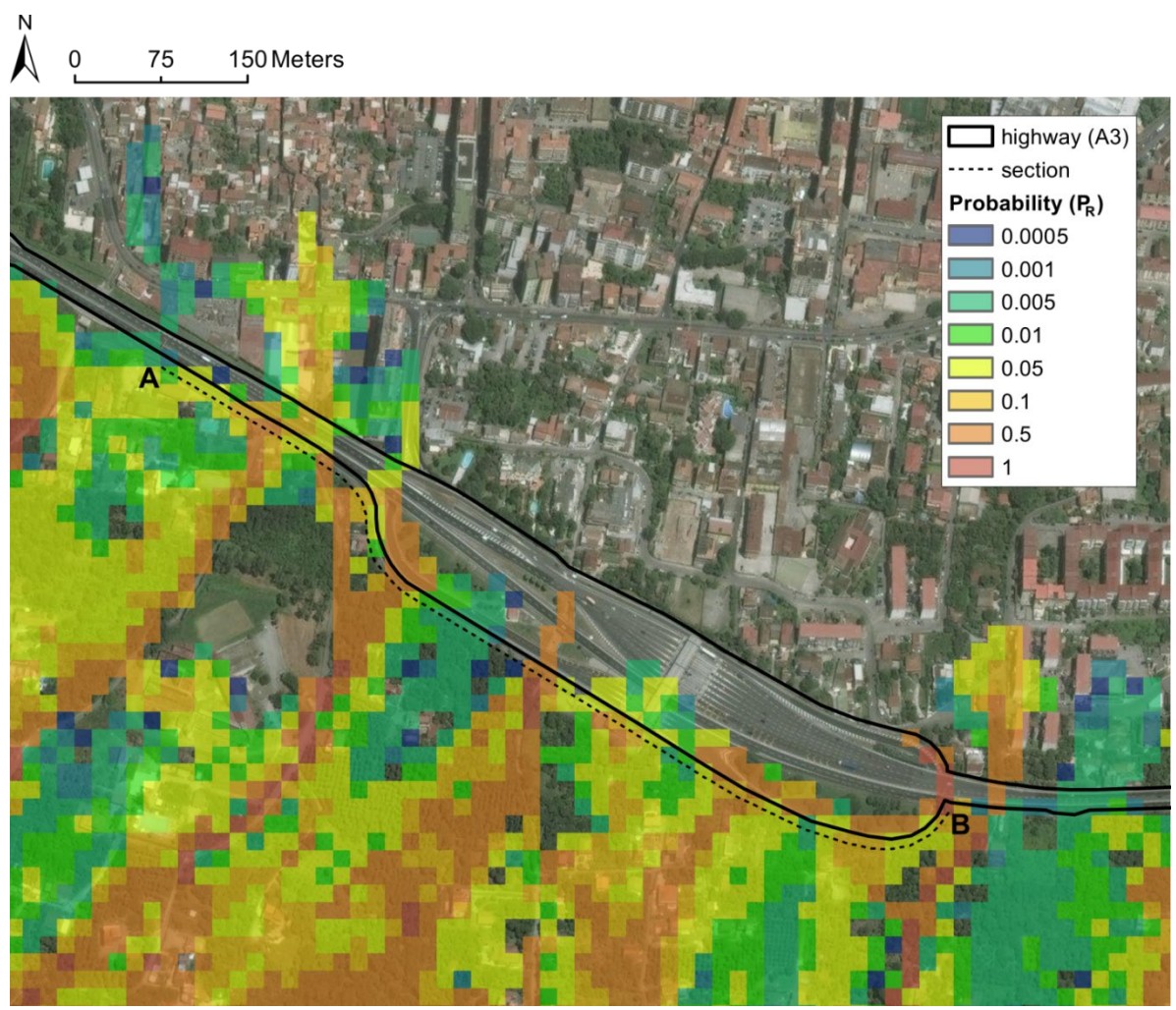

Figure 12. Spatial distribution of reach probability at infrastructure scale and indication of section A-B

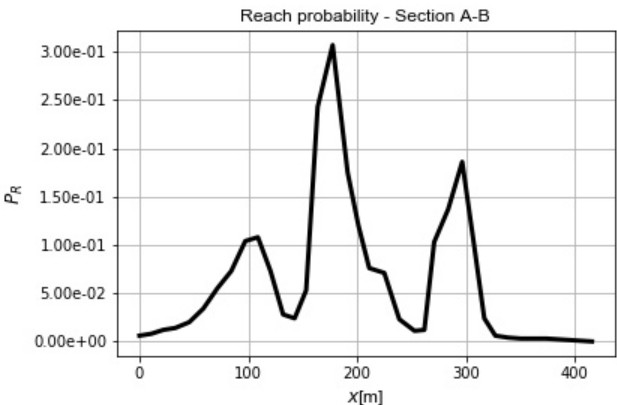

Figure 13. Reach probability along the A-B section (Figure 12) of the A3 motorway (point A is located at x=0)

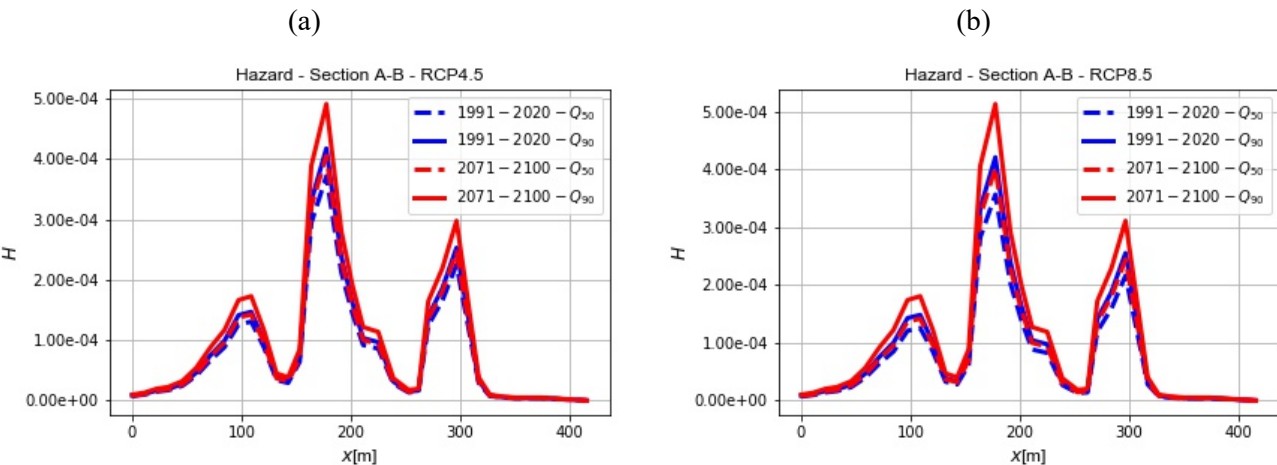

Figure 14. Landslide hazard for section A-B, calculated for time intervals 1991-2020 and 2071-2100 and for quantiles $Q_{50}$ and $Q_{90}$: (a) RCP4.5; and (b) RCP8.5