# Peer review of "Temporal evolution of flow-like landslide hazard for a road infrastructure in the Municipality of Nocera Inferiore (southern Italy) under the effect of climate change"

_Natural Hazards and Earth System Sciences, 2017_

## Referee Comment (RC1) · Anonymous Referee #1 · 5 Jan 2018

General Comments This paper attempt to evaluate landslide hazard under the climate change. Because there is a possibility that the climate change will affect frequency of landslide hazard in the future, topic of this paper is very important. Statistical approaches, which are used in this study, is nice for evaluation of landslide hazard in the long-term future, because there are many uncertainties that physical models cannot overcome. However, there are some weakness in this paper.

1) Linkage between triggering probability and reach probability are expressed as equation (1) (p.4). However, the H (hazard probability?) are not calculated in this paper. Analyses of the triggering probability and the reach probability have been done sepa-

rately, and never been linked together. Therefore, I felt that this paper is composed of two different studies.

2) Although relationship between the climate change and the triggering probability are presented in chapter 5, there is no analysis on influence of the climate change on the reach probability. Because one of the most important aspect of this study is estimation of landslide risk under the climate change (as noted in 1. Introduction), effect of the climate change to the reaching probability is needed in this paper. This problem occurs because of the poor linkage between analysis of triggering probability and reach probability as I pointed out in the comment 1).

3) Statements in discussion parts (latter half in chapter 5, section 6.2) and the concluding section (chapter 7) are mostly about case example in the study site. General findings applicable to other areas are limited.

4) There are many assumptions in the analysis of this study. I agree that this kind of works need assumptions, because it is hard to obtain detailed data needed for the analysis. In addition, there are many uncertainties as authors discussed in the chapters 1 and 7. However, when the authors set important assumptions, explanations on reasonability of the assumption (or discussion on limitations in the assumption) are needed. See specific comments.

5) Locations (or characteristics) of source area and runout area of previous landslides are not shown in this paper. Such information is important when we consider if the assumption in this paper is realistic or not. The landslide histories can be used to verify result of the prediction.

Specific comments

pg.3, line 22 "(a) hyper-concentrated flows, which are. . .as debris avalanches"

Is there any difference in rainfall threshold and runout distance amongst these three landslide types? Many previous studies have reported that travel distance (and slope

angle) of landslides and debris flows are variable amongst different topography and different types of the mass movement. Gavan Hunter, Robin Fell (2003) Canadian Geotechnical Journal, 40, 1123-1141 R J Fannin, M P Wise (2001) Canadian Geotechnical Journal, 38, 982-994 C Scheidland, D Rickenmann (2010) Earth Surf. Process. Landforms, 2010, 35, 157–173 J Corominas (1996) Canadian Geotechnical Journal, 33, 260-271 In chapter 6, authors did not distinguish landslide types when they estimate the reach provability. Therefore, they assumed that the landslide type does not affect runout characteristics. Difference (and similarity) in the runout characteristics amongst landslide types is helpful for readers to consider reasonability of the assumption. Similar things can be said to the landslide triggering condition.

pg.4, line 4 "resolution of 15x15 m"

This resolution is larger than that recommended by Horton et al. (2013) NHESS. Why do you think this grid size is sufficient for estimation of the reach probability? It is hard to understand from the statements in chapter 6.

pg. 4 line 8 Equation (1)

H in the equation (1) can be given by the triggering probability multiplied by the reach probability. In my understanding, triggering probability indicates the probability of occurrence of one landslide in the entire analysis area (if only one landslide occurred during each rainfall event in table 2). However, if the reach probability was multiplied by the triggering probability, it means that landslides simultaneously occur at all of source areas during one rainfall event. Maybe I am misunderstanding the method, but detailed explanation is needed to prevent misunderstanding.

pg. 5, line 6 "The inventory of landslide events was. . .the Regional Civil Protection"

What kind of data do the reports include? Landslide timing? Locations of source area and runout area?

pg.6, line 1 "In the present study, climate simulations included in EURO-CORDEX

multi-model ensemble at 0.11' (approximately 12 km) are considered under the RCP4.5 and RCP8.5 scenarios as described in Table 3."

Difference in the triggering probability between RCP4.5 and RCP8.5 (Fig. 7) are based on the difference in the rainfall characteristics between the two scenarios. However, rainfall characteristics of the two scenarios are not explained in this paper. I suggest to explain difference in the rainfall characteristics between the two scenarios.

pg. 6, line 7 "Landslide triggering probability was estimated. . .and the 59-day rainfall ðİŽ¡59."

Why one-day rainfall and 59-day rainfall were used in the analysis? Rainfall intensity and duration are generally used in this kind of analysis (e.g., Berti et al., 2012). Berti et al., Journal of Geophysical Research, 117, F04006, 2012

pg. 7, line 10 "More specifically, Fig. 7a shows. . . variation for both scenarios"

This sentence is repetition of the Figure caption. I suggest to remove this sentence.

pg. 9, line 3-5 "In this work, source areas were identified. . ."

In this study, zero order basin and current failure areas are considered as source areas. Does this assumption agree with location of previous landslides in this area? Although this hypothesis are briefly explained in the next sentence, detailed explanations are needed, because setting of the source area is one of the most important factor controlling runout areas.

pg. 9, line 12 "An angle of reach of 4° was calibrated based on the geomorphological information (i.e., the extension of the slope fan deposition). . .."

"Extension of the slope fan deposition" is the maximum travel distance of the landslide. Do you mean all landslides possibly reach the end of fan deposition if there is no limitation by the flow velocity? As many papers have reported, landslide runout distance is variable depending on the landslide volume and landslide type (e.g., Corominas, 1996,

CGJ). I afraid that the "angle of reach" in this study overestimates the reach probability.

pg. 9, section 6.2

Results and discussion are mostly about spatial distribution of the runout area. However, the runout area is mainly controlled by "angle of reach" and "maximum velocity", which are arbitrary set by authors. Therefore, results and discussion of probability is more important than the runout area. I suggest that authors add results and discussion on the probability.

Table 1

Coordinate of weather station at Castellammare di Stabia should be expressed by degree-minute-second.

Table 2

Please note the date of March 2005 event in Gragnano.

Table 2

How many landslides occurred during each event?

Fig. 1

A scale and a north arrow are needed.

Fig. 2

Does the area named M. Albino correspond to the area of Fig. 8? Please clarify.

Fig. 3

I think the area surrounded by the red line is the highway. Please note that in the figure caption.

Fig. 4

"Estimation of landslide triggering probability for RCP 4.5 and RCP8.5 scenarios" and "Estimation and mapping of reach probability" have been done in this study. However, three items at the bottom of the flow chart have not been done. Therefore, it is hard for me to image procedure in the last part of this flowchart.

Fig. 10

In the x-axis, the value "0" may indicate location of the point A. Please clarify.
* * *

---

## Referee Comment (RC2) · Anonymous Referee #2 · 31 Jan 2018

The manuscript "Temporal evolution of landslide hazard for a road infrastructure in the Municipality of Nocera Inferiore, Italy, under the effect of climate change" tries to develop a statistical model aimed to the probability of occurrence of a landslide using Bayesian approach and apply it to a small study area located in Italy. Moreover the authors evaluate the impact of possible climatic change on such probability through the use of EURO-CORDEX climatic models.

Even if the topic can be considered important, the paper seems to be not completely developed and there are some weaknesses in the methods used and in the presentation of the results.

[Figure]

1) Definition of the statistical model: the authors use two variables derived by rainfall measurements as proxy of landslide triggering: 1-day rainfall and 59-day rainfall. The choice of these variables is briefly mentioned by the authors (page 6 – lines 7-9) but is totally unclear. Since the choice of the proxy variables is essential in the definition of probability of landslides triggering, this part deserves more space and more details.

2) The authors define the hazard as the product between probability of landslide triggering and the reach probability (which, in my opinion, can be defined in a more appropriate way). The authors affirm that that probability of triggering is only related to the rainfall (parameters??) and is assumed constant over the space while only the reach probability depends on the morphology and is spatially variable. I think that these assumptions are very questionable and affect the entire research. Moreover even if the authors show this definition of hazard, it is not applied and assessed in the manuscript (no figure shows hazard maps). The figure 4 (flow chart of the study is not in agreement with the results presented in the manuscript).

3) The results of the triggering probability in the future (2071-2100) are questionable as well if it is inserted in the context of IPCC AR5 results for the Mediterranean area. IPCC AR5 forecasts a strong reduction of the rainfall for this area at seasonal and annual scale. Since the authors use as landslide triggering proxy precipitation at 59 days and 1 day, the increase of landslide triggering probability seems to be a little bit controversial. The reader has no tools to try to understand the reason of this behavior.

4)The authors provide no assessment of the performance of the landslide triggering method.

5) There are different assumptions (sometimes very important, especially on the derivation of the different probabilities which compose the hazard), which are not explained with the proper details and which are very questionable. The authors should add more details each time they introduce an assumption trying to explain the possible consequences of such assumptions.

6) In the section 6.1 the stop of run-out routing is related to the exceeding of a velocity parameter and it is not clear the role of this parameter in the method used by the authors. Also other concepts, as the persistence function, are not properly explained by the authors.

7) I'm no English mother tongue but some parts of the paper are very hard to read and to understand – I suggest the use of English native speaker to re-read the paper and correct it.

---

## Author Comment (AC1) · 22 Mar 2018

The Authors wish to thank the Reviewer for his/her comments. Please see below the detailed responses by comment number.

Comment 1.1: Linkage between triggering probability and reach probability are expressed as equation (1) (p.4). However, the H (hazard probability?) are not calculated in this paper. Analyses of the triggering probability and the reach probability have been done separately, and never been linked together. Therefore, I felt that this paper is composed of two different studies.

[Figure]

Response: Triggering probability and reach probability are distinct parameters which depend from different factors and which are computed separately as shown in the paper. They are only linked in that they both appear in the hazard model. Triggering probability defines the likelihood of the triggering of at least one landslide in the study area as a consequence of the occurrence of given thresholds of cumulative rainfall. Reach probability describes the probability that any spatial location will be reached by a moving soil mass, assuming that landslides have been triggered. Additional text will be included in Section 3 "Method of analysis" providing more precise definitions of triggering probability and reach probability, and their conceptual significance in the overall hazard estimation approach. A fully worked computation of hazard for the case study will be conducted in the revised version according to the model proposed in Eq. (1).

Comment 1.2: Although relationship between the climate change and the triggering probability are presented in chapter 5, there is no analysis on influence of the climate change on the reach probability. Because one of the most important aspect of this study is estimation of landslide risk under the climate change (as noted in 1. Introduction), effect of the climate change to the reaching probability is needed in this paper. This problem occurs because of the poor linkage between analysis of triggering probability and reach probability as I pointed out in the comment 1).

Response: Reach probability is not related to climate change, as it parameterizes the probability of spatial occupation during landslide runout, assuming that triggering has occurred in one or more potential source areas. Reach probability only depends on terrain factors. Climate change is related to triggering probability through the probability of exceedance of the 1-day and 59-day cumulative rainfall thresholds. Additional text will be included in Section 3 "Method of analysis" providing more precise definitions of triggering probability and reach probability, and their conceptual significance in the overall hazard estimation approach.

Comment 1.3: Statements in discussion parts (latter half in chapter 5, section 6.2) and

the concluding section (chapter 7) are mostly about case example in the study site. General findings applicable to other areas are limited.

Response: Following the Sarno event (1998), many investigations on landslides affecting pyroclastic covers on slopes of Campania Region have been carried out. These have stressed the important role played by local geomorphological conditions for landslide occurrence: cover depth and stratigraphy regulated by distance from eruptive centers and wind directions and magnitude during the eruptions, exposition, slope angle, bottom hydraulic conditions. These studies highlight the importance of avoiding generalizations regarding predictive models of Early Warning systems (e.g. I-D curves). Under such constraints, in this work, we prefer considering a unique geomorphological context as detected by Picarelli et al. (2008) . The macro zoning is intended mainly regulating the proxies. Nevertheless, the framework is totally replicable in other contexts where proxies implying the same stages are already known.

Comment 1.4: There are many assumptions in the analysis of this study. I agree that this kind of works need assumptions, because it is hard to obtain detailed data needed for the analysis. In addition, there are many uncertainties as authors discussed in the chapters 1 and 7. However, when the authors set important assumptions, explanations on reasonability of the assumption (or discussion on limitations in the assumption) are needed. See specific comments.

Response: Thank you for the suggestion. We will refer to the "Specific Comments" section in attempting to address the Reviewer's inputs.

Specific comments

Comment 1.5: Locations (or characteristics) of source area and runout area of previous landslides are not shown in this paper. Such information is important when we consider if the assumption in this paper is realistic or not. The landslide histories can be used to verify result of the prediction.

Response: Thank you for the suggestion. Please see below: Source areas: Source areas were identified by means of the official geo-morphological map of the "Campania Centrale" River Basin Authority (PSAI 2015) and coincide with the union of the 1) "zero order basin" (ZOB) and the 2) actual "niche/failure" areas. Figure 8 shows the perimeter that envelopes the two areas above mentioned. In the revised version, we will add a geo-morphological map with the two distinct components and specifications (and references) to clarify the assumptions. Runout areas: In the above-mentioned map, the traces of previous landslides obtained from the official landslides inventory of the "Campania Centrale" River Basin Authority (PSAI 2015) will be shown.

Comment 1.6: - pg.3, line 22 "(a) hyper-concentrated flows, which are…as debris avalanches" Is there any difference in rainfall threshold and runout distance amongst these three landslide types? Many previous studies have reported that travel distance (and slope angle) of landslides and debris flows are variable amongst different topography and different types of the mass movement. Gavan Hunter, Robin Fell (2003) Canadian Geotechnical Journal, 40, 1123-1141 R J Fannin, M P Wise (2001) Canadian Geotechnical Journal, 38, 982-994 C Scheidland, D Rickenmann (2010) Earth Surf. Process. Landforms, 2010, 35, 157–173 J Corominas (1996) Canadian Geotechnical Journal, 33, 260-271 In chapter 6, authors did not distinguish landslide types when they estimate the reach provability. Therefore, they assumed that the landslide type does not affect runout characteristics. Difference (and similarity) in the runout characteristics amongst landslide types is helpful for readers to consider reasonability of the assumption. Similar things can be said to the landslide triggering condition.

Response: The landslide catalogue used for retrieving triggering probability primarily refer to debris flow in channelized or open slopes (see De Vita and Piscopo, 20022). The landslide types considered in that study are: (1) "channelized debris flows, which can be generated by slope failure in ZOB areas (Dietrich et al. 1986; Cascini et al. 2008)" and (2) un-channelized debris flows, which are locally triggered on open-slopes areas propagating as debris avalanches. We will specify it in the revised version. Just

only one un-channelized event (March 2005) occurred in Nocera (Pagano et al. 2010; Rianna et al. 2014). The "niche/failure" areas of this specific event are considered as source areas in the runout analysis. The event/runout characteristics of the above-mentioned two landslide types can be significantly different; nevertheless, the same calibration parameter set (reach angle, velocity) seems to satisfy enough both event conditions.

Comment 1.7: pg.4, line 4 "resolution of 15x15 m" - This resolution is larger than that recommended by Horton et al. (2013) NHESS. Why do you think this grid size is sufficient for estimation of the reach probability? It is hard to understand from the statements in chapter 6.

Response: Horton et al. (2013) stated the following: "a 10m DEM resolution as a good compromise between processing time and quality of results. However, valuable results have still been obtained on the basis of lower quality DEMs with 25m resolution". A variety of DTM resolutions were tested for the case study. We opine that the adopted resolution adequately represents the surface morphology (simply comparing – numerically and by expert judgment – the DTM with the real current morphological shape of the areas – the resolution represents with a good accuracy the channelized shape and the fan areas). This assessment will be described in greater detail in the revised version.

Comment 1.8: pg. 4 line 8 Equation (1) - H in the equation (1) can be given by the triggering probability multiplied by the reach probability. In my understanding, triggering probability indicates the probability of occurrence of one landslide in the entire analysis area (if only one landslide occurred during each rainfall event in table 2). However, if the reach probability was multiplied by the triggering probability, it means that landslides simultaneously occur at all of source areas during one rainfall event. Maybe I am misunderstanding the method, but detailed explanation is needed to prevent misunderstanding.

Response: This study replicates the hypotheses and glossary introduced by Berti et al. (2012) regarding the implications and possible limitations of the Bayesian approach to quantifying landslide triggering probability empirically. Regarding the specific aspect discussed by the reviewer, this study adopts the modelling hypothesis by Berti et al. (2012) by which multiple landslides are counted as one single event. Hence, the Bayesian method presented in the paper quantifies the probability of occurrence of the event (defined as "at least one landslide in the proximity area"). Reach probability as defined and calculated in the study is consistent with this definition, as the results obtained are calculated as the superposition of all possible runout paths from all landslides potentially occurring from all source areas. Hazard as calculated using the above hypotheses is a conservative, upper-bound estimate related to a specific rainfall scenario involving specific values of 1-day and 59-day cumulative rainfall. These hypotheses, along with additional insights into conceptual background of the Bayesian approach to landslide triggering estimation, will be presented more explicitly in the revised version.

Comment 1.9: pg. 5, line 6 "The inventory of landslide events was…the Regional Civil Protection" - What kind of data do the reports include? Landslide timing? Locations of source area and runout area?

Response: As reported in the paper, multiple sources are used for reconstructing the inventory. De Vita and Piscopo (2002)2, for example, report for events in the same geomorphological context the cumulative rainfall values inducing the events on time spans up to 60 days. Vallario (2000) provides brief descriptions about the events (also for the other natural hazards affecting the Region) including also the number of fatalities and injured. "Event Reports", drafted by the Regional Civil Protection, report exhaustive descriptions about the weather patterns inducing the triggering event and the main consequences for the affected communities.

Comment 1.10: pg.6, line 1 "In the present study, climate simulations included in EURO-CORDEX multi-model ensemble at 0.11' (approximately 12 km) are considered

under the RCP4.5 and RCP8.5 scenarios as described in Table 3." Differences in the triggering probability between RCP4.5 and RCP8.5 (Fig. 7) are based on the difference in the rainfall characteristics between the two scenarios. However, rainfall characteristics of the two scenarios are not explained in this paper. I suggest to explain difference in the rainfall characteristics between the two scenarios.

Response: The trends of maximum daily precipitation and 59 days cumulative values under the two RCPs will be reported in the revised version of the paper in terms of ensemble mean value and related spread. This will allow a clear insight into future evolution of weather forcing parameters which are of interest for landslide occurrence.

Comment 1.11: pg. 6, line 7 "Landslide triggering probability was estimated...and the 59-day rainfall." - Why one-day rainfall and 59-day rainfall were used in the analysis? Rainfall intensity and duration are generally used in this kind of analysis (e.g., Berti et al., 2012). Berti et al., Journal of Geophysical Research, 117, F04006, 2012.

Response: Thank you for the suggestion. In the revised version, we will clarify the rationale behind this choice. Several works (De Vita and Piscopo, 2002 ; Fiorillo & Wilson, 2004 ; Pagano et al., 2010 ; Napolitano et al., 2016 ; Comegna et al., 2017 ;Reder et al., 2018 ) stressed the prominent role of antecedent precipitations for landslide occurrence in pyroclastic covers. However, the effective length of such window is highly dependent from local conditions. In this perspective, for the same geomorphological context, De Vita and Piscopo (2002) use again 59 days. Preliminary analyses were conducted using a number of proxies in the calibration of the Bayesian approach developed in the paper. Such analyses showed that 1-day and 59-day rainfall could be confidently used. The results of preliminary analyses involving other proxies will be briefly mentioned in the revised version.

Comment 1.12: pg. 7, line 10 "More specifically, Fig. 7a shows... variation for both scenarios" - This sentence is repetition of the Figure caption. I suggest to remove this sentence.

[Figure]

Response: Thank you for the suggestion. This will be done in the revised version.

Comment 1.13: pg. 9, line 3-5 "In this work, source areas were identified..." - In this study, zero order basin and current failure areas are considered as source areas. Does this assumption agree with location of previous landslides in this area? Although this hypothesis are briefly explained in the next sentence, detailed explanations are needed, because setting of the source area is one of the most important factor controlling runout areas.

Response: Thank you for the suggestion. This will be done in the revised version.

Comment 1.14: pg. 9, line 12 "An angle of reach of 4_ was calibrated based on the geomorphological information (i.e., the extension of the slope fan deposition)..." - "Extension of the slope fan deposition" is the maximum travel distance of the landslide. Do you mean all landslides possibly reach the end of fan deposition if there is no limitation by the flow velocity? As many papers have reported, landslide runout distance is variable depending on the landslide volume and landslide type (e.g., Corominas, 1996, CGJ). I afraid that the "angle of reach" in this study overestimates the reach probability.

Response: Yes, runout distance is variable depending on the landslide volume and landslide type and all landslides cannot reach the end of fan deposition. In this work, reach probability was evaluated considering a "paroxysmal" event, based on the official geomorphological characteristics and the official "high" and "extremely high" hazard areas. The assessment by process-based modelling at a large scale (no single event/flow or slope) is generally difficult due to the complex nature of the phenomenon, the variability of local controlling factors, and the uncertainty in modelling parameters. We used a simplified approach that is not highly parameter-dependent. Maps and specification will be added in the revised version to clarify these aspects.

Comment 1.15: pg. 9, section 6.2 - Results and discussion are mostly about spatial distribution of the runout area. However, the runout area is mainly controlled by "angle of reach" and "maximum velocity", which are arbitrary set by authors. Therefore, results

and discussion of probability is more important than the runout area. I suggest that authors add results and discussion on the probability.

Response: Thank you for the suggestion. This will be done in the revised version. Comment 1.16: Table 1 - Coordinate of weather station at Castellammare di Stabia should be expressed by degree-minute-second.

Response: Thank you for the suggestion. This will be done in the revised version.

Comment 1.17: Table 2 - Please note the date of March 2005 event in Gragnano.

Response: Thank you for the suggestion. This will be done in the revised version.

Comment 1.18: Table 2 - How many landslides occurred during each event?

Response: For the three towns, the date reported in Table 2 refers to single main event; in this perspective, when, for a same weather patterns, landslide events have been observed in two towns (for example, 10-11 January 1997), they are reported as different occurrences.

Comment 1.19: Fig. 1 - A scale and a north arrow are needed.

Response: Thank you for the suggestion. This will be done in the revised version.

Comment 1.20: Fig. 2 - does the area named M. Albino correspond to the area of Fig. 8? Please clarify.

Response: Thank you for the input. We will introduce in the caption of Figure 8 the indication of the area shown in Figure 2

Comment 1.21: Fig. 3 - I think the area surrounded by the red line is the highway. Please note that in the figure caption.

Response: Thank you. We will add this significant detail in the Figure's caption

Comment 1.22: Fig. 4 - "Estimation of landslide triggering probability for RCP 4.5 and RCP8.5 scenarios" and "Estimation and mapping of reach probability" have been done

in this study. However, three items at the bottom of the flow chart have not been done. Therefore, it is hard for me to image procedure in the last part of this flowchart.

Response: The terminal phases of the procedure will be addressed more explicitly through the completion of the case study example. This will be achieved through additional text, figures and tables.

Comment 1.23: Fig. 10 - In the x-axis, the value "0" may indicate location of the point A. Please clarify.

Response: Thank you. We will add this significant detail in the Figure's caption

---

## Author Comment (AC2) · 22 Mar 2018

The Authors wish to thank the Reviewer for his/her comments. Please see below the detailed responses by comment number.

Comment 2.1 Definition of the statistical model: the authors use two variables derived by rainfall measurements as proxy of landslide triggering: 1-day rainfall and 59-day rainfall. The choice of these variables is briefly mentioned by the authors (page 6 – lines 7-9) but is totally unclear. Since the choice of the proxy variables is essential in the definition of probability of landslides triggering, this part deserves more space and more details.

[Figure]

Response: Thank you for the suggestion. In the revised version, we will clarify the rationale behind this choice. Several works (De Vita and Piscopo, 2002 ; Fiorillo & Wilson, 2004 ; Pagano et al., 2010 ; Napolitano et al., 2016 ; Comegna et al., 2017 ;Reder et al., 2018 ) stressed the prominent role of antecedent precipitations for landslide occurrence in pyroclastic covers. However, the effective length of such window is highly dependent from local conditions. In this perspective, for the same geomorphological context, De Vita and Piscopo (2002) use again 59 days. Preliminary analyses were conducted using a number of proxies in the calibration of the Bayesian approach developed in the paper. Such analyses showed that 1-day and 59-day rainfall could be confidently used. The results of preliminary analyses involving other proxies will be briefly mentioned in the revised version.

Comment 2.2 The authors define the hazard as the product between probability of landslide triggering and the reach probability (which, in my opinion, can be defined in a more appropriate way). The authors affirm that that probability of triggering is only related to the rainfall (parameters??) and is assumed constant over the space while only the reach probability depends on the morphology and is spatially variable. I think that these assumptions are very questionable and affect the entire research. Moreover even if the authors show this definition of hazard, it is not applied and assessed in the manuscript (no figure shows hazard maps). The figure 4 (flow chart of the study is not in agreement with the results presented in the manuscript).

Response: The definition of hazard as the product between triggering probability and reach probability reflects the quantitative approach based on conditional probability. If hazard can be defined as the probability that a specific spatial location can be reached as a consequence of the occurrence of a given combination of values of the control parameters (in this case, 1-day and 59-days cumulative rainfall), then it is conceptually correct to define this as the product of "the probability that an event is triggered given the occurrence of a given combination of value of the control parameters" (triggering probability) and " the probability that a specific spatial location is reached in the course

of the event's runout" (reach probability). The probability of triggering in this specific study is investigated in terms of rainfall parameters (the aforementioned 1-day and 59-days cumulative rainfall). Triggering probability is assumed constant within the study area since the database which is used to develop the Bayesian method refers to the area itself. As detailed in a similar study by Berti et al. (2012), the quantitative output of empirical methods such as the one developed in the paper implicitly accounts for the spatial variability (if any) of rainfall characteristics within the area. Additional text will be provided in the revised version to explain more explicitly the issue of spatial variability of triggering factors as related to the scale of analysis and to the specific study area. Triggering probability as defined in the study does not parameterize the component of landslide susceptibility related to terrain characteristics. This component is considered in the reach probability term. The numerical method used to model landslide runout addresses the spatial variability of the terrain-related susceptibility component through the specification of hillslope areas prone to slide (source areas) extracted by official mapped geo-morphological and hazard data. The source areas have been chosen in relation to the specific analyzed landslide type (i.e. ZOB and actual "niche/failure" for channelized debris flow; actual "niche/failure" for un-channelized debris flow). The terminal phases of the procedure will be addressed more explicitly through the completion of the case study example. This will be achieved through additional text, figures and tables.

Comment 2.3 The results of the triggering probability in the future (2071-2100) are questionable as well if it is inserted in the context of IPCC AR5 results for the Mediterranean area. IPCC AR5 forecasts a strong reduction of the rainfall for this area at seasonal and annual scale. Since the authors use as landslide triggering proxy precipitation at 59 days and 1 day, the increase of landslide triggering probability seems to be a little bit controversial. The reader has no tools to try to understand the reason of this behavior.

Response: The findings about future trends of daily and cumulative precipitations are

provided by CORDEX initiative performing a multi-model ensemble on Europe at very high resolution. The main results provided by EURO-CORDEX are included in IPCC Assessment Reports in attempting to improve and localize the general results provided by Global Climate Models. Moreover, in our view, our results are not controversial. Indeed, the reductions assessed for cumulative values at seasonal scale can be linked to higher retention capability of atmosphere potentially induced by global warming; the same mechanism could induce increases in occurrence and magnitude of heavy rainfall events due to high amount of "precipitable" water in atmosphere.

Comment 2.4 The authors provide no assessment of the performance of the landslide triggering method.

Response: The method developed in the paper is a predictive method which looks into the future evolution of landslide hazard in probabilistic terms. Regarding the estimation of triggering probability, the Bayesian approach employed in the paper is inspired by the one proposed by Berti et al. (2012). This study, similarly to the former one, inherently incorporates past information about the empirical relationship between triggering factors and occurrence of events in the Bayesian formulation; more specifically, in the likelihood probability term. Thus, from a quantitative point of view, the Bayesian approach explicitly accounts for past evidence. The estimation of reach probability is based on the numerical modeling of landslide runout for events originating in potential source areas.

Comment 2.5 There are different assumptions (sometimes very important, especially on the derivation of the different probabilities which compose the hazard), which are not explained with the proper details and which are very questionable. The authors should add more details each time they introduce an assumption trying to explain the possible consequences of such assumptions.

Response: Thank you for the suggestion. In the revised version, we will clearly identify assumptions in the different sections. Moreover, we will clarify the points specifically

identified by the two Reviewers.

Comment 2.6 In the section 6.1 the stop of run-out routing is related to the exceeding of a velocity parameter and it is not clear the role of this parameter in the method used by the authors. Also other concepts, as the persistence function, are not properly explained by the authors.

Response: Thank you for the suggestion. In the revised version, we will explain these aspects.

Comment 2.7: I'm no English mother tongue but some parts of the paper are very hard to read and to understand – I suggest the use of English native speaker to re-read the paper and correct it.

Response: The revised version will be carefully re-read to improve its readability. A native English speaker could support us in this task.

---

## Author Response (AR1)

The Authors wish to thank the Reviewers for their valuable comments. The manuscript was thoroughly reviewed and improved through:

- Significant revision of the text in all sections of the paper
- More detailed explanations of the conceptual and procedural approaches in Sections 3, 5 and 6
- Significant reorganization of Sections 5 and 6, addressing landslide occurrence probability and reach probability, respectively
- Addition of Section 7, addressing hazard calculation
- Revision of existing figures and tables
- Addition of 4 figures (Figures 5, 9, 10 and 14)
- Addition of 1 table (Table 4)
- Editing of syntax and glossary

Please see the detailed responses by reviewer and comment number.

**RESPONSES TO REVIEWER 1:**

**General comments**

**Comment 1.1:**

Linkage between triggering probability and reach probability are expressed as equation (1) (p.4). However, the H (hazard probability?) are not calculated in this paper. Analyses of the triggering probability and the reach probability have been done separately, and never been linked together. Therefore, I felt that this paper is composed of two different studies.

*Response:*

Section 3 "Method of analysis", Section 5 "Landslide occurrence probability" and Section 6 "Reach probability" have been thoroughly reviewed and significantly modified to clarify the operational approach employed in the study. A fully worked computation of hazard for the case study has been included in a new section (Section 7 "Calculation of hazard") in the revised version. Figure 4 has been modified for consistency with the revised glossary.

**Comment 1.2:**

Although relationship between the climate change and the triggering probability are presented in chapter 5, there is no analysis on influence of the climate change on the reach probability. Because one of the most important aspect of this study is estimation of landslide risk under the climate change (as noted in 1. Introduction), effect of the climate change to the reaching probability is needed in this paper. This problem occurs because of the poor linkage between analysis of triggering probability and reach probability as I pointed out in the comment 1).

*Response:*

Reach probability is not related to climate change, as it parameterizes the probability of spatial occupation during landslide runout, assuming that triggering has occurred in one or more potential source areas. Reach

probability only depends on terrain factors. Climate change is related to landslide occurrence probability through the probability of exceedance of the 1-day and 59-day cumulative rainfall thresholds. Section 3 "Method of analysis" has been thoroughly reviewed and significantly modified to clarify the operational approach employed in the study. More precise definitions of triggering probability, occurrence probability and reach probability have been included in the revised version, as well as more detailed explanations of their interrelations and individual attributes (e.g., dependency (or lack thereof) from climate change, spatial variability, temporal variability, etc.

**Comment 1.3:**

Statements in discussion parts (latter half in chapter 5, section 6.2) and the concluding section (chapter 7) are mostly about case example in the study site. General findings applicable to other areas are limited.

*Response:*

The Authors are grateful to the Reviewer stressing a potentially critical point in the text. The following text has been added to the conclusions (Section 8):

"Campanian pyroclastic covers are characterized by several specific features (high porosity, significant water retention capacity, intermediate saturated hydraulic conductivities) playing a relevant role for landslide triggering (e.g. role of antecedent precipitations or persistency/magnitude of potential triggering event). Moreover, stratigraphic details as the actual grain size distribution, the presence of pumice lenses or the depth of pyroclastic deposits regulated by the distance from the eruptive centers and wind direction/magnitude during the eruptions make complex also generalisations within the same Campania Region. Nevertheless, the framework developed for the pyroclastic covers on the North side of the Monti Lattari (where Nocera Inferiore is located) appears easily transferable to other contexts where precipitation observations and details about the timing of landslide events are available. Similarly, the climate simulation chain follows the state-of-the-art for analysis of impacts potentially induced by climate changes. Finally, the estimated increases in hazard result consistent with those reported in several works investigating the variation in frequency of landslide events in coarse grained soils (Gariano & Guzzetti, 2016)."

**Comment 1.4:**

There are many assumptions in the analysis of this study. I agree that this kind of works need assumptions, because it is hard to obtain detailed data needed for the analysis. In addition, there are many uncertainties as authors discussed in the chapters 1 and 7. However, when the authors set important assumptions, explanations on reasonability of the assumption (or discussion on limitations in the assumption) are needed. See specific comments.

*Response:*

The text has been thoroughly reviewed, and all attempts have been made to ensure that assumptions and hypotheses underlying are duly explained and clarified.

**Specific comments**

**Comment 1.5:**

Locations (or characteristics) of source area and runout area of previous landslides are not shown in this paper. Such information is important when we consider if the assumption in this paper is realistic or not. The landslide histories can be used to verify result of the prediction.

*Response:*

Source areas were identified by means of the official geo-morphological map of the "Campania Centrale" River Basin Authority (PSAI 2015) and coincide with the union of the 1) "zero order basin" (ZOB) and the 2) actual "niche/failure" areas. A new figure (Figure 10)  a new map with the geo-metrological elements of interest and the runout of previous landslides obtained from the official landslides inventory of the "Campania Centrale" River Basin Authority (PSAI 2015) is included in the revised version. Figure 11 shows the perimeter (locations) that envelopes the two areas above mentioned (source areas).

**Comment 1.6:**

pg.3, line 22 "(a) hyper-concentrated flows, which are…as debris avalanches"

Is there any difference in rainfall threshold and runout distance amongst these three landslide types? Many previous studies have reported that travel distance (and slope angle) of landslides and debris flows are variable amongst different topography and different types of the mass movement. Gavan Hunter, Robin Fell (2003) Canadian Geotechnical Journal, 40, 1123-1141 R J Fannin, M P Wise (2001) Canadian Geotechnical Journal, 38, 982-994 C Scheidland, D Rickenmann (2010) Earth Surf. Process. Landforms, 2010, 35, 157–173 J Corominas (1996) Canadian Geotechnical Journal, 33, 260-271 In chapter 6, authors did not distinguish landslide types when they estimate the reach provability. Therefore, they assumed that the landslide type does not affect runout characteristics. Difference (and similarity) in the runout characteristics amongst landslide types is helpful for readers to consider reasonability of the assumption.
Similar things can be said to the landslide triggering condition.

*Response:*

The following text was added in Section 6.2: "The landslide catalogue used for retrieving triggering probability primarily refer to debris flow in channelized or open slopes (see De Vita and Piscopo, 2002[2]). The landslide types considered in that study are: (1) "channelized debris flows, which can be generated by slope failure in ZOB areas (Dietrich et al. 1986; Cascini et al. 2008)" and (2) un-channelized debris flows, which are locally triggered on open-slopes areas propagating as debris avalanches. We specified it in the revised version. Just only one un-channelized event (March 2005) occurred in Nocera (Pagano et al. 2010; Rianna et al. 2014). The "niche/failure" areas of this specific event are considered as source areas in the runout analysis. The event/runout characteristics of the above-mentioned two landslide types can be significantly different; nevertheless, the same calibration parameter set (reach angle, velocity) seems to satisfy enough both event conditions."

**Comment 1.7:**

pg.4, line 4 "resolution of 15x15 m"

This resolution is larger than that recommended by Horton et al. (2013) NHESS. Why do you think this grid size is sufficient for estimation of the reach probability? It is hard to understand from the statements in chapter 6.

*Response:*

Horton et al. (2013) stated the following: "a 10m DEM resolution as a good compromise between processing time and quality of results. However, valuable results have still been obtained on the basis of lower quality DEMs with 25m resolution".

A variety of DTM resolutions were tested for the case study. We opine that the adopted resolution adequately represents the surface morphology (simply comparing – numerically and by expert judgment – the DTM with the real current morphological shape of the areas – the resolution represents with a good accuracy the channelized shape and the fan areas) confirming the Horton et al. (2013) observations (cfr. Introduction)

**Comment 1.8:**

pg. 4 line 8 Equation (1)

H in the equation (1) can be given by the triggering probability multiplied by the reach probability. In my understanding, triggering probability indicates the probability of occurrence of one landslide in the entire analysis area (if only one landslide occurred during each rainfall event in table 2). However, if the reach probability was multiplied by the triggering probability, it means that landslides simultaneously occur at all of source areas during one rainfall event. Maybe I am misunderstanding the method, but detailed explanation is needed to prevent misunderstanding.

*Response:*

This study replicates the hypotheses and glossary introduced by Berti et al. (2012) regarding the implications and possible limitations of the Bayesian approach to quantifying landslide triggering probability empirically. Regarding the specific aspect discussed by the reviewer, this study adopts the modelling hypothesis by Berti et al. (2012) by which multiple landslides are counted as one single event. Hence, the Bayesian method presented in the paper quantifies the probability of occurrence of the event (defined as "at least one landslide in the proximity area"). Reach probability as defined and calculated in the study is consistent with this definition, as the results obtained are calculated as the superposition of all possible runout paths from all landslides potentially occurring from all source areas. Hazard as calculated using the above hypotheses is a conservative, upper-bound estimate related to a specific rainfall scenario involving specific values of 1-day and 59-day cumulative rainfall. These hypotheses, along with additional insights into conceptual background of the Bayesian approach to landslide triggering estimation, have been included and explained explicitly in the revised version.

**Comment 1.9:**

pg. 5, line 6 "The inventory of landslide events was…the Regional Civil Protection"

What kind of data do the reports include? Landslide timing? Locations of source area and runout area?

*Response:*

The following text was added to Section 4.2 "Landslide inventory":

"The multiple sources used for reconstructing the inventory provide quite different details. De Vita and Piscopo (2002), for example, report the cumulative rainfall values inducing the events on time spans up to 60 days for

events in the same geomorphological context. Vallario (2000) provides brief descriptions about the events (also for the other natural hazards affecting the Region) including the number of fatalities and injured. "Event Reports", drafted by the Regional Civil Protection, contain exhaustive descriptions about the weather patterns inducing the triggering event and the main consequences for the affected communities."

**Comment 1.10:**

pg.6, line 1 "In the present study, climate simulations included in EURO-CORDEX multi-model ensemble at 0.11' (approximately 12 km) are considered under the RCP4.5 and RCP8.5 scenarios as described in Table 3."

Differences in the triggering probability between RCP4.5 and RCP8.5 (Fig. 7) are based on the difference in the rainfall characteristics between the two scenarios. However, rainfall characteristics of the two scenarios are not explained in this paper. I suggest to explain difference in the rainfall characteristics between the two scenarios.

*Response:*

The following text was added to Section 4.3 "Climate projections" and a new figure (Figure 5) was included.

"In Figure 5, the variations expected in monthly cumulative values (5a) and maximum daily precipitations (5b) are displayed assuming 1981-2010 as reference period and splitting the period 2010-2100 in three 30-year periods. More specifically, the upper part of Figure 5a shows the expected variations in monthly cumulative variations for RCP 4.5 (continuous line) and RCP8.5 (hatched line) as returned by bias-corrected projections in the short-term (green; 2011-2040 vs 1981-2010), medium-term (blue; 2041-2070 vs 1981-2010) and long-term (red; 2071-2100 vs 1981-2010). The bottom part of Figure 5a shows the observed annual cycle of monthly cumulative precipitations (in mm). Figure 5b shows the mean values of maximum daily precipitations in the reference observed period (1982-2009) and projected on short-term (green: 2011-2040 vs 1981-2010), medium-term (blue: 2041-2070 vs 1981-2010) and long-term (red: 2071-2100 vs 1981-2010). Filled and dashed bars correspond to results for RCP4.5 and RCP8.5, respectively.

 The ensemble mean values from EURO-CORDEX optimally overlaps the actual values (data not displayed) for the same time span. Concerning future time periods, reductions up to 45% (under RCP8.5) are expected in the summer season. In this perspective, the decreases are mainly regulated by the severity of concentration scenarios. Values generally lower than the current ones are also estimated in spring (approximately -10%) and in the first part of autumn (approximately -5%). These predictions are characterized by a fluctuating signal. An increase is expected in the remaining seasons, with few exceptions (i.e., short term 2011-2040 under RCP4.5). Higher increases could exceed 20% in November and 15% in January. These evolutions could primarily induce variations in the timing of landslide events affecting pyroclastic covers in the area. Such events tend to occur especially in the second part of winter (or first part of spring) following the increase in antecedent precipitations. On the contrary, the likelihood of occurrence reduces during autumn and in the first part of winter. It is also worth noting that the expected increase in temperature (not taken into account in this approach) could lead to a higher atmospheric evaporative demand and, thus, to lower values of soil water content within the pyroclastic covers. Regarding precipitation triggering events, the variations in maximum daily precipitation are displayed in Figure 4b. Under both scenarios, increases with respect the reference value (about 90 mm/day) ranging from 5 and 15% for "mid-way" scenario and as high as 20% are expected under RCP8.5 for the intermediate time horizon."

**Comment 1.11:**

pg. 6, line 7 "Landslide triggering probability was estimated…and the 59-day rainfall."

Why one-day rainfall and 59-day rainfall were used in the analysis? Rainfall intensity and duration are generally used in this kind of analysis (e.g., Berti et al., 2012). Berti et al., Journal of Geophysical Research, 117, F04006, 2012.

*Response:*

The following text was added to Section 5.1 "Landslide occurrence probability calculation method":

"Several studies have stressed the prominent role of antecedent precipitations for landslide occurrence in pyroclastic covers: De Vita and Piscopo (2002) used 59-day rainfall for the same geomorphological context; Napolitano et al., (2016) defined different Intensity-Duration (I-D) rainfall thresholds for dry and wet seasons for the Sarno area. Comegna et al. (2017) assessed through a statistical framework that effective precipitation period for the Monti Lattari area could be 3 months long. Fiorillo & Wilson (2004) suggested a simplified approach to evaluate the attainment of soil moisture states which could act as landslide triggering factors. Pagano et al. (2010), interpreting the 2005 landslide events in Nocera Inferiore, suggested that antecedent precipitations, should be considered at least 4-months long for those events. Reder et al. (2018) stressed the role of soil-atmosphere water exchanges during the entire hydrological year, accounting also for the effect of evaporation losses. They also stated that the effective length of effective antecedent precipitation window is highly dependent from local conditions: cover depth, pumice lenses, bottom hydraulic conditions."

**Comment 1.12:**

pg. 7, line 10 "More specifically, Fig. 7a shows… variation for both scenarios"

This sentence is repetition of the Figure caption. I suggest to remove this sentence.

*Response:*

The text has been significantly modified in the context of the reorganization of Section 5.

**Comment 1.13:**

pg. 9, line 3-5 "In this work, source areas were identified…"

In this study, zero order basin and current failure areas are considered as source areas. Does this assumption agree with location of previous landslides in this area? Although this hypothesis are briefly explained in the next sentence, detailed explanations are needed, because setting of the source area is one of the most important factor controlling runout areas.

*Response:*

This assumption agree with location of previous landslides in this area. In order to support this assumption, a new figure (Figure 10) showing the map with the geo-metrological elements of interest and the runout of previous landslides obtained from the official landslides inventory of the "Campania Centrale" River Basin Authority (PSAI 2015) is included in the revised version. Observed landslides originated in ZOB and/or in "niche/failure" areas.

**Comment 1.14:**

pg. 9, line 12 "An angle of reach of 4_ was calibrated based on the geomorphological information (i.e., the extension of the slope fan deposition)."

"Extension of the slope fan deposition" is the maximum travel distance of the landslide. Do you mean all landslides possibly reach the end of fan deposition if there is no limitation by the flow velocity? As many papers have reported, landslide runout distance is variable depending on the landslide volume and landslide type (e.g., Corominas, 1996, CGJ). I afraid that the "angle of reach" in this study overestimates the reach probability.

*Response:*

Runout distance is indeed variable, depending on the landslide volume and landslide type. Not all landslides reach the end of fan deposition. In this study, reach probability was estiamted considering a "paroxysmal" event, based on the official geomorphological characteristics (Fan and detrital fan) and the official hazard areas (See the new Figure 10). Due to the large-scale of the assessment and the complexity of the analyzed phenomena, a not highly parameter-dependent approach was deliberately chosen The assessment by process-based modelling at a large scale (no single event/flow or slope) is generally difficult due to the complex nature of the phenomenon, the variability of local controlling factors, and the uncertainty in modelling parameters.

**Comment 1.15:**

pg. 9, section 6.2 - Results and discussion are mostly about spatial distribution of the runout area. However, the runout area is mainly controlled by "angle of reach" and "maximum velocity", which are arbitrary set by authors. Therefore, results and discussion of probability is more important than the runout area. I suggest that authors add results and discussion on the probability.

*Response:*

Reach probability has been explained in greater detail in Sections 3, 6, 7 and 8 in the revised version.

**Comment 1.16:**

Table 1 - Coordinate of weather station at Castellammare di Stabia should be expressed by degree-minute-second.

*Response:*

Table 1 has been modified as suggested.

**Comment 1.17:**

Table 2 - Please note the date of March 2005 event in Gragnano.

*Response:*

Table 2 has been modified as suggested.

**Comment 1.18:**

Table 2 - How many landslides occurred during each event?

*Response:*

For the three towns, the date reported in Table 2 refers to single main event; in this perspective, when, for a same weather patterns, landslide events have been observed in two towns (for example, 10-11 January 1997), they are reported as different occurrences.

**Comment 1.19:**

Fig. 1 - A scale and a north arrow are needed.

*Response:*

Figure 1 has been modified as suggested.

**Comment 1.20:**

Fig. 2 -  Does the area named M. Albino correspond to the area of Fig. 8? Please clarify.

*Response:*

The caption of Figure 11 (previously Figure 8) has been modified as follows:

"Figure 11. Spatial distribution of reach probability at hillslope scale; the area corresponds to the box named "Mt. Albino" in Figure 2"

**Comment 1.21:**

Fig. 3 - I think the area surrounded by the red line is the highway. Please note that in the figure caption.

*Response:*

The caption of Figure 3 has been modified as follows:

"Figure 3. Infrastructure-scale view of the study area with the A3 Salerno-Reggio Calabria motorway (boundaries marked in red)"

**Comment 1.22:**

Fig. 4 - "Estimation of landslide triggering probability for RCP 4.5 and RCP8.5 scenarios" and "Estimation and mapping of reach probability" have been done in this study. However, three items at the bottom of the flow chart have not been done. Therefore, it is hard for me to image procedure in the last part of this flowchart.

*Response:*

A fully worked computation of hazard for the case study has been included in a new section (Section 7) in the revised version. The discussion of results has been extended in greater detail in Section 8 (formerly Section 7).

**Comment 1.23:**

Fig. 10 - In the x-axis, the value "0" may indicate location of the point A. Please clarify.

*Response:*

The caption of Figure 13 (previously Figure 10) has been modified as follows:

"Figure 13. Reach probability along the A-B section of the A3 motorway (point A is located at x=0)"

**RESPONSES TO REVIEWER 2:**

**Comment 2.1**

Definition of the statistical model: the authors use two variables derived by rainfall measurements as proxy of landslide triggering: 1-day rainfall and 59-day rainfall. The choice of these variables is briefly mentioned by the authors (page 6 – lines 7-9) but is totally unclear. Since the choice of the proxy variables is essential in the definition of probability of landslides triggering, this part deserves more space and more details.

*Response:*

The following text was added to Section 5.1 "Landslide occurrence probability calculation method":

"Several studies have stressed the prominent role of antecedent precipitations for landslide occurrence in pyroclastic covers: De Vita and Piscopo (2002) used 59-day rainfall for the same geomorphological context; Napolitano et al., (2016) defined different Intensity-Duration (I-D) rainfall thresholds for dry and wet seasons for the Sarno area. Comegna et al. (2017) assessed through a statistical framework that effective precipitation period for the Monti Lattari area could be 3 months long. Fiorillo & Wilson (2004) suggested a simplified approach to evaluate the attainment of soil moisture states which could act as landslide triggering factors. Pagano et al. (2010), interpreting the 2005 landslide events in Nocera Inferiore, suggested that antecedent precipitations, should be considered at least 4-months long for those events. Reder et al. (2018) stressed the role of soil-atmosphere water exchanges during the entire hydrological year, accounting also for the effect of evaporation losses. They also stated that the effective length of effective antecedent precipitation window is highly dependent from local conditions: cover depth, pumice lenses, bottom hydraulic conditions."

**Comment 2.2**

The authors define the hazard as the product between probability of landslide triggering and the reach probability (which, in my opinion, can be defined in a more appropriate way). The authors affirm that that probability of triggering is only related to the rainfall (parameters??) and is assumed constant over the space while only the reach probability depends on the morphology and is spatially variable. I think that these assumptions are very questionable and affect the entire research. Moreover even if the authors show this definition of hazard, it is not applied and assessed in the manuscript (no figure shows hazard maps). The figure 4 (flow chart of the study is not in agreement with the results presented in the manuscript).

*Response:*

Section 3 "Method of analysis", Section 5 "Landslide occurrence probability" and Section 6 "Reach probability" have been thoroughly reviewed and significantly modified to clarify the operational approach employed in the study. A fully worked computation of hazard for the case study has been included in a new section (Section 7 "Calculation of hazard") in the revised version. Figure 4 has been modified for consistency with the revised glossary.

**Comment 2.3**

The results of the triggering probability in the future (2071-2100) are questionable as well if it is inserted in the context of IPCC AR5 results for the Mediterranean area. IPCC AR5 forecasts a strong reduction of the rainfall for this area at seasonal and annual scale. Since the authors use as landslide triggering proxy precipitation at 59 days and 1 day, the increase of landslide triggering probability seems to be a little bit controversial. The reader has no tools to try to understand the reason of this behavior.

*Response:*

The following text and a new figure (Figure 5) were added to Section4.3 "Climate projections":

"In Figure 5, the variations expected in monthly cumulative values (5a) and maximum daily precipitations (5b) are displayed assuming 1981-2010 as reference period and splitting the period 2010-2100 in three 30-year periods. More specifically, the upper part of Figure 5a shows the expected variations in monthly cumulative variations for RCP 4.5 (continuous line) and RCP8.5 (hatched line) as returned by bias-corrected projections in the short-term (green; 2011-2040 vs 1981-2010), medium-term (blue; 2041-2070 vs 1981-2010) and long-term (red; 2071-2100 vs 1981-2010). The bottom part of Figure 5a shows the observed annual cycle of monthly cumulative precipitations (in mm). Figure 5b shows the mean values of maximum daily precipitations in the reference observed period (1982-2009) and projected on short-term (green: 2011-2040 vs 1981-2010), medium-term (blue: 2041-2070 vs 1981-2010) and long-term (red: 2071-2100 vs 1981-2010). Filled and dashed bars correspond to results for RCP4.5 and RCP8.5, respectively.

 The ensemble mean values from EURO-CORDEX optimally overlaps the actual values (data not displayed) for the same time span. Concerning future time periods, reductions up to 45% (under RCP8.5) are expected in the summer season. In this perspective, the decreases are mainly regulated by the severity of concentration scenarios.  Values generally lower than the current ones are also estimated in spring (approximately -10%) and in the first part of autumn (approximately -5%). These predictions are characterized by a fluctuating signal. An increase is expected in the remaining seasons, with few exceptions (i.e., short term 2011-2040 under RCP4.5). Higher increases could exceed 20% in November and 15% in January. These evolutions could primarily induce variations in the timing of landslide events affecting pyroclastic covers in the area. Such events tend to occur especially in the second part of winter (or first part of spring) following the increase in antecedent precipitations. On the contrary, the likelihood of occurrence reduces during autumn and in the first part of winter. It is also worth noting that the expected increase in temperature (not taken into account in this approach) could lead to a higher atmospheric evaporative demand and, thus, to lower values of soil water content within the pyroclastic covers. Regarding precipitation triggering events, the variations in maximum daily precipitation are displayed in Figure 4b. Under both scenarios, increases with respect the reference value (about 90 mm/day) ranging from 5 and 15% for "mid-way" scenario and as high as 20% are expected under RCP8.5 for the intermediate time horizon."

**Comment 2.4**

The authors provide no assessment of the performance of the landslide triggering method.

*Response:*

The method developed in the paper is a predictive method which looks into the future evolution of landslide hazard in probabilistic terms. Regarding the estimation of triggering probability, the Bayesian approach employed in the paper is inspired by the one proposed by Berti et al. (2012). This study, similarly to the former one, inherently incorporates past information about the empirical relationship between triggering factors and

occurrence of events in the Bayesian formulation; more specifically, in the likelihood probability term. Thus, from a quantitative point of view, the Bayesian approach to the estimation of triggering probability explicitly accounts for past evidence. These aspects have been clarified and discussed in Section 5.1 and Section 8.

**Comment 2.5**

There are different assumptions (sometimes very important, especially on the derivation of the different probabilities which compose the hazard), which are not explained with the proper details and which are very questionable. The authors should add more details each time they introduce an assumption trying to explain the possible consequences of such assumptions.

*Response:*

The text has been thoroughly reviewed, and all attempts have been made to ensure that assumptions and hypotheses underlying are duly explained and clarified.

**Comment 2.6**

In the section 6.1 the stop of run-out routing is related to the exceeding of a velocity parameter and it is not clear the role of this parameter in the method used by the authors. Also other concepts, as the persistence function, are not properly explained by the authors.

*Response:*

The approach used (Horton et al. 2013) may result in improbable runout distances in steep catchments due to unrealistic energy amounts reached during the propagation. To keep the energy within reasonable values, the method allows to define a maximum limit to ensure not to exceed realistic velocities. The persistence function Gamma (2000) aims at reproducing the behavior of inertia and weights the flow direction based on the change in direction with respect to the previous direction. The runout routine used in this work and the other concepts referenced in Section 6.1 are thoroughly explained in Horton et al. (2013) and are thus not explained in detail in our manuscript.

[revised manuscript text omitted]
 C̶o̶n̶f̶i̶n̶i̶n̶g̶ ̶t̶h̶e̶ ̶s̶t̶u̶d̶y̶ ̶j̶u̶s̶t̶ ̶t̶o̶on a 400-meter a̶ ̶p̶a̶r̶tstretch of the infrastructure (e̶.̶g̶.̶,̶ ̶f̶r̶o̶m̶ ̶A̶ ̶t̶o̶ ̶B̶from point A to point B

20 in Figure 12), the runout values to be considered in the risk assessment should be taken along the section A-B (F̶i̶g̶. ̶1̶0̶Figure 1̶12). H̶a̶z̶a̶r̶d̶ ̶c̶a̶n̶ ̶b̶e̶ ̶c̶a̶l̶c̶u̶l̶a̶t̶e̶d̶ ̶d̶i̶r̶e̶c̶t̶l̶y̶ ̶f̶o̶r̶ ̶a̶ ̶g̶i̶v̶e̶n̶ ̶y̶e̶a̶r̶ ̶a̶n̶d̶ ̶R̶C̶P̶ ̶s̶c̶e̶n̶a̶r̶i̶o̶ ̶b̶y̶ ̶a̶p̶p̶l̶y̶i̶n̶g̶ ̶E̶q̶. ̶(̶1̶)̶. ̶The results shown in F̶i̶g̶. ̶1̶0̶Figure 1̶13 attest t̶ofor the marked spatial variability of reach probability (̶a̶n̶d̶,̶ ̶t̶h̶e̶r̶e̶f̶o̶r̶e̶,̶ ̶o̶f̶ ̶h̶a̶z̶a̶r̶d̶)̶ along the investigated section of the A3 motorway infrastructure.

**7̶7 Calculation of hazard**

25 Once occurrencet̶r̶i̶g̶g̶e̶r̶i̶n̶g̶ probability and reach probability have been estimated as illustrated a̶b̶o̶v̶ein Section 5 and Section 6, respectively, it is possible to calculate hazard using Eq. (1). Hazard is temporally variable because t̶r̶i̶g̶g̶e̶r̶i̶n̶g̶occurrence probability i̶s̶ ̶e̶x̶p̶l̶i̶c̶i̶t̶l̶y̶ ̶m̶o̶d̶e̶l̶l̶e̶d̶ ̶a̶s̶ ̶b̶e̶i̶n̶g̶displays temporal variability t̶e̶m̶p̶o̶r̶a̶l̶l̶y̶ ̶v̶a̶r̶i̶a̶b̶l̶e̶ as a consequence of climate change as shown in Section 5.3. Reach probability is assumed to be temporally invariant as it is deterministically related to terrain morphology. This entails that the reach probability outputs obtained in Section 6.2 v̶a̶l̶i̶d̶i̶t̶y̶ ̶o̶f̶ ̶t̶h̶i̶s̶ ̶s̶t̶u̶d̶y̶ ̶i̶s̶ ̶l̶i̶m̶i̶t̶e̶dare valid

30 t̶oonly for the current terrain morphology. Should significant variations in s̶u̶c̶hterrain morphology occur, for instance, in case of the occurrence of landslide events, reach probability would need to be reassessed as described in Section 6.1.̶.

Fig. 12: as old Figure 9

Fig. 13: as old Figure 10To complete the flowchart shown in Figure 4, an example calculation of hazard is provided for the section A-B. Figure 14 shows the spatially and temporally variable hazard profile for time intervals 1991-2020 and 2071-2100, for both quantiles $Q_{50}$ and $Q_{90}$ and for RCP4.5 and RCP 8.5. The occurrence probability values used to multiply the reach probability values shown in Figure 13 are taken from Table 4.

**8 Concluding remarks**

This paper has illustrated an innovative methodology for the quantitative estimation of rainfall-induced landslide hazard. An example application of the proposed method was conducted for a short section of a motorway. Despite the limited extension of the study area, the results displayed a marked temporal and spatial variability of hazard. The temporal variability of hazard is a consequence of climate change as parameterized through quantitative projections for concentration scenarios RCP4.5 and RCP8.5. Significant temporal variability was assessed for both concentration scenarios. The considerable spatial variability resulting from the case study stems from the spatial variability of reach probability as modelled in the runout analysis.

The calculation of occurrence probability, specifically in the triggering probability calculation phase, relies on a Bayesian approach which replicates the one provided by Berti et al. (2012). This study replicates the hypotheses and glossary introduced by these Researchers, and shares Berti et al. (2012) regarding the implications, and possible limitations of the Bayesiansuch approach to quantifying landslide triggering probability empirically. For instance, tThe 
[revised manuscript text omitted]

[Figure]

**Geo-morhological elements (of interest)**

| | | | |
|---|---|---|---|
| �(brown) | Zero Order Basin | ▢ (light blue) | Detrital fan |
| ▢ (green) | Niche/failure areas | ▢ (blue) | Fan |
| ▢ | Landslides inventory | | |

**Hazard**

| | | | |
|---|---|---|---|
| ▢ (red) | Very high | ▢ (yellow) | Moderate |
| ▢ (orange) | High | ▢ (green) | Low |

Figure 10. Geo-morphological map of the "Campania Centrale" River Basin Authority (PSAI 2015)

[Figure]

Figure 11. Spatial distribution of reach probability at hillslope scale; the area corresponds to the box named "Mt. Albino" in Figure 2

[Figure]

Figure 12. Spatial distribution of reach probability at infrastructure scale and indication of section A-B

[Figure]

Figure 13. Reach probability along the A-B section of the A3 motorway (point A is located at x=0)

[Figure]

(a)                                                         (b)

Figure 14. Landslide hazard for section A-B, calculated for time intervals 1991-2020 and 2071-2100 and for quantiles $Q_{50}$ and $Q_{90}$: (a) RCP4.5; and (b) RCP8.5

---

## Referee Report (RR1)

**RE:** NHESS 2017 416R1          Uzielli et al.  Temporal evolution of landslide hazard for a road infrastructure in the Municipality of Nocera Inferiore, southern Italy, under the effect of climate change.

**Overview**

The authors improved the paper. However, there is a point to be exploited yet. The hazard they estimate, results from a probability calculation. Really, according to the Technical Italian Law, the estimation of hazard is quantitative and is usually obtained after considering three scenarios of different return period. The hazard the authors determine, seems a susceptibility hazard rather than a hazard. Therefore, the authors in the introduction should defining the hazard they are investigating after introducing the other hazard approaches. In this sense for clarity, they should also add that the "quantitative hazard" is determined in different way by using models able to simulate both deposition and entrainment (Deangeli, 2008, Rosatti and Begnudelli, 2013; Frank et al. 2015, Stancanelli et al. 2015, Cuomo et al. 2016, and Gregoretti et al. 2018)

The following are the detailed comments to the unclear sentences.

At page 1 line 20, what does it mean "concentration scenario"?

At page 1 line 25, while (conversely)????? The authors should choose one.

At page 1 line 25 there are two commas

At page 2, lines 21-22 not clear sentence

At page 2 line 27: references missing for the other studies

At page 3 line 7: The meaning of the sentence "Flow-like landslide…..." does not match the manuscript: the reach probability is the second step after the triggering probability for estimating the hazard. It is not an added value, something more as the sentence shows.

At page 3, line 25: what is it ZOB? Please define it.

At page 3 lines 24-27: the difference between channelized and un-channelized is not clear.

At page 4, line 2   "….estimate of hazard and its mapping" is better

At page 4 the sentence  "The study area was modelled into the GIS software through a digital terrain model (DTM) having a resolution of 15x15 m." is rather unclear

Cuomo, S., Pastor, M., Capobianco, V., Cascini, L., (2016). Modelling the space- time bed entrainment for flow-like landslide. Engineering Geology. 212, 10-20. doi10.10116/j.enggeo.2016.07.011

Deangeli, C., (2008). CLaboratory granular flows generated by slope failures. Rock Mechanics and Rock Engineering. 41(1), 199-217.

Frank, F., McArdell, B.W., Huggel C., Vieli. A., (2015). The importance of entrainment and bulking on debris flow runout modeling: examples from the Swiss Alps. Nat. Hazards Earth Syst. Sci. 15, 2569-2583, doi:10.5194/nhess-15-2569-2015

Gregoretti, C., Degetto, M. Bernard, M. Boreggio M. (2018) The debris flow occurred at Ru Secco Creek, Venetian Dolomites, on 4 August 2015: analysis of the phenomenon, its characteristics and reproduction by models. Frontier in Earth Sciences, doi: 10.3389/feart.2018.00080 (accepted for publication)

Rosatti, G., Begnudelli, L., 2013. Two dimensional simulations of debris flows over mobile beds: enhancing the TRENT2D model by using a well-balanced generalized Roe-type solver. Comput. Fluids 71, 179–185. http://dx.doi.org/10.1016/j.compfluid2012.10.006.

Stancanelli L. M., Foti E., (2015). A comparative assessment of two different debris flows propagation approaches - blind simulation on a real debris flow event. Natural Hazard Earth System Science, 15, 375-746. doi:10.5194/nhess-15-375-2015.

---

## Author Response (AR2)

**REVIEWER 1**

**Overview**

The authors propose a method to compute the probability, varying in time according to the climate change, that a location is invested by a landslide occurred elsewhere upstream. The writer carefully read until subsection 1.2 and gave a quick look to the remainder because the manuscript is not easily readable: many sentences are ill structured and entirely or partially appear unclear. The writing is not fluid, somewhere is confused, and somewhere it seems translated italian. The writer suggests the authors to ask for the help of colleagues with a good knowledge of the topic and English writing. In addition, there are also four main deficiencies:

*Response: We wish to thank the Reviewer for the insights provided to improve the manuscript. We have reviewed the text and hope it will prove to be more easily readable following the revision.*

1) The author associate to the word "hazard" the spatial varying probability of a location to be invested by a moving mass resulting from a landslide occurred elsewhere upstream. Such association seems arbitrary. The writer suggests something like "hazard function" or similar. Moreover, the "reach probability" could confuse the reader because reach is also a segment of a channel/river. Perhaps impacting probability could be better: the authors could find a better name for naming the probability of a location to be invested by such a phenomenon.

*Response: We thank the Reviewer for raising the matter. Nevertheless, we consider that the terms used in the paper could be consistent with current specialist technical glossary.*

*The term "hazard", intended as the probability of occurrence of an event of a given magnitude in a given spatial location, is used consistently with the globally adopted definitions provided by UNISDR (United Nations Office for Disaster Risk Reduction), ISSMGE TC304 and JTC-1 (Joint Technical Committee 1) among others. The approach pursued in the paper is a zonal, site-level analysis of the spatial variability of hazard. The term "reach probability" is commonly used to define the frequency of a moving mass impacting a specific spatial location. References making use of this term include, among many others, the recent following contributions:*

*Nappi, M., Budetta, P. Lombardi, G. and Minotta C.: Rockfall run-out estimate comparing empirical and trajectographic approaches. In: Landslide Science and Practice: Volume 6: Risk Assessment, Management and Mitigation, Springer, 177-182, 2006.*

*Zhang, S., Zhang, L.M., Peng, M., Zhang, L.L., Zhao, H.F. and Chen. H.X.: Assessment of risks of loose landslide deposits formed by the 2008 Wenchuan earthquake. Natural Hazards and Earth System Sciences, 12, 1381-1392, 2012.*

*Fell, R.: Human induced landslides. In: Landslides and Engineered Slopes. Experience, Theory and Practice (Ed. Aversa, Cascini, Picarelli, Scavia), CRC Press, 2016.*

*Valaguss, A., Frattini, P. and Crosta, G.B: Quantitative probabilistic hazard analysis of earthquake-induced rockfalls. In: Landslide Science for a Safer Geoenvironment: Volume 3: Targeted Landslides (Ed. Sassa, Canuti, Yin), 213-218, 2014.*

2) The authors deal with landslide runout and sometimes refer to debris flows. In principle, they are not the same thing. Some of the landside transformed into a debris flow? If this is the case, it should written at the beginning specifying the debris flow type (in present case perhaps some muddy or viscous debris flow: please see figure 1.11 of Takahashi, 2007). At page 3 the phenomena are hyper-concentrated flows, channelized debris flows and un-channelized debris flows. However, table 2 these phenomena are classified as landslide. This does not seem correct. In the text it can ben derived that all the landslide phenomena transformed in flows: hyper-concentrated flows, channelized debris flows and un-channelized debris flows. In such a case it should be correct use the diction debris flow runout. The writer suggests the authors to introduce a specific characterization of the phenomena occurring in the site they are studying.

***Response:*** *We thank the Reviewer for requesting clarification about this important aspect. We have revised the to address this comment.*

*In accordance with Hutchinson (2004), we used the term "flow-like landslide" to generally define "flow-like movements" in granular material, such as pyroclastic cover. In the Manuscript P3L28-P4L3, where we define "hyper-concentrated flows", "channelized and un-channelized debris lows", we have added a specific characterization showing in brackets for "hyper-concentrated flows" the definition "flows in transition from mass transport to mass movement" and for "channelized and un-channelized debris-flows" respectively channelized and un-channelized "flow-like mass movements".*

*The terms generally used in the paper will be "flow-like landslide" or "flow-like movements". Relating to "runout" term, we coupled it with flow-like landslide (e.g. flow-like landslide runout).*

3) The authors assume that the landslide occurrence and reach probability are distinct parameters (line 12 of page 3). What does it mean? Which is the link between probability and parameter? They assume that these two probabilities are independent: equation (1) deals with a joint probability. Really, the probability of location to be routed by a moving mass seems linked to the probability of the mass moving occurrence. Therefore, the authors should carefully explain this assumption.

***Response:*** *The paper defines occurrence probability as follows (P4L8-13): "Occurrence probability defines the likelihood of the occurrence of at least one event in the study area as a consequence of the attainment of given thresholds of cumulative rainfall and of the likelihood of triggering given the occurrence of such thresholds." Occurrence does not imply that all spatial locations are reached during runout; rather, it entails that a movement is initiated in at least one source area. However, occurrence is a necessary condition for runout to occur, i.e., for any spatial location to be impacted during the runout phase. In our conditional probability approach, hazard is given, for any spatial location in the area, by the product of the probability of occurrence of the movement and the probability that the occurred movement actually reaches that spatial location. The probabilities are independent because the occurrence probability depends on the triggering phenomenon and reach probability depends on the runout model. As stated in the paper, (P4L14-23) "Occurrence probability is partly related to the likelihood of triggering given the attainment of specific rainfall thresholds, which is assumed to be an inherent, time-invariant attribute of the area, and partly related to climate change through the probability of exceedance of such rainfall thresholds as described in Section 5. Reach probability is not related to climate change, as it parameterizes the probability of spatial occupation during runout, assuming that triggering has occurred. Reach probability depends solely on terrain factors. Occurrence and triggering probabilities are related to rainfall parameters and,*

*thus, are assumed to be spatially invariant and uniform for the entire area, while reach probability depends on geomorphological factors, and is thus cell-specific and spatially variable within the area."*

4) The authors apply and test the proposed methodology to a reach of the Motorway A3 threatened by the channels incising Mount Albino. The routing modeling at the base of the probability reach calculations is carried out using Flow-R. This model it is a susceptibility model based on empirical parameters that does not consider the volume of the flowing mass. Therefore, the writer suggests the use, for only one channel threatening the motor way reach, of a physically based model (Deangeli, 2008, Armanini et al. 2009, Frank et al. 2015, Stancanelli et al. 2015, Cuomo et al. 2016, and Gregoretti et al. 2016) at the purpose to test the simulations carried out by using flow-R . This comparison should help to increase the reliability of the simulations.

**Response:** *We thank the Reviewer for the suggestions. In Section 6.2 - Reach probability outputs, the text has been modified to: "An angle of reach of 4° was calibrated based on the geo-morphological information (i.e., the extension of the slope fan deposition) and the official hazard maps of the Landslide Risk Management Plan of the River Basin Authority (PSAI, 2015) shown in Figure 10, considering a "paroxysmal" event."*

[Figure]

**Geo-morhological elements (of interest)**

| | |
|---|---|
| Zero Order Basin | Detrital fan |
| Niche/failure areas | Fan |
| Landslides inventory | |

**Hazard**

| | |
|---|---|
| Very high | Moderate |
| High | Low |

*Figure 10. Geo-morphological (left) and Hazard (right) maps of the Landslide Risk Management Plan of the River Basin Authority (PSAI, 2015). The above-mentioned hazard maps are also the result*

*of previous physically-based modeling studies performed or supervised by the River Basin Authority. Anyway, we greatly appreciate the suggestions of the Reviewer (that will be surely taken into account in the coming developments of the work) but we honestly think that adding a new section (e.g. "reliability of the reach simulations") could divert the focus from the "core" of the work (climate change effect, Bayesian approach, etc.); furthermore, it is likely to lengthen a part of the paper (runout analysis) that, in the previous revisions, reviewers suggested to shorten it (and we did it).*

The writer also suggests to add "Southern" before Italy in the title.

*Response: the title has been modified according to the reviewer's suggestion.*

The following are the detailed comments to the unclear sentences.

**Abstract**

At line 19, please substitute "following" with "during the"

*Response: The text has been modified according to the Reviewer's suggestion.*

**Introduction**

At page 1 line 27, please substitute "While probability distributions of "shallow" uncertainties in outcomes are "reasonably well known" " with "Shallow uncertainties are associated to reasonably well know probabilities of outcomes (Stein…..), while (conversely) deep uncertainties are associated to…….

*Response: The text has been modified according to the Reviewer's suggestion.*

At page 2, line 3 eliminate one of the two issues

*Response: we do not know if we have properly understood the Reviewer's comment; nonetheless, we would prefer to report the three different items for sake of consistency with Hallegatte (2012).*

At page 2, line 6 please substitute "related to climate change" with "in a changing climate"

*Response: The text has been modified according to the Reviewer's suggestion.*

At page 2, line 19 please substitute "investigations performed in them" with "relative investigations" and eliminate "For example".

*Response: The text has been modified according to the Reviewer's suggestion. The term "respective" was used in lieu of "relative".*

At page 2, line 21 please substitute "such differences induce variations in rainfall patterns recognized as effective for slope failure" with "Therefore, the critical rainfall pattern inducing slope failure varies according to these differences.

*Response: The text has been modified according to the Reviewer's suggestion.*

At page 2, line 22 please substitute "For these reasons, while daily weather forcing data have been found to result in better assessments for the Cervinara and Nocera Inferiore test cases, sub-daily data have been found to improve the quality of assessments for the Ravello test case." with "Indeed in some locations (Cervinara and Nocera Inferiore), the assessed weather forcing has a daily duration, while in other locations (Ravello) it has sub-daily duration.

*Response: The text has been modified according to the Reviewer's suggestion.*

At page 2, line 24 please substitute "Consequently, daily observations modified according to projected anomalies (Damiano & Mercogliano 2013) or daily data provided by climate simulations subjected to statistical bias correction are used in the former cases, while a stochastic approach is adopted with bias-corrected data to provide assessments at hourly scale for the latter." With "Consequently two different approaches are followed for (the scope of such a data elaboration) based on the considered duration. The former relative to daily durations is ........., the latter relative to sub-daily durations is..............."

*Response: Following the Reviewer's suggestions, the text has been modified as: "Consequently, two different approaches are followed for (the scope of such a data elaboration) based on the considered duration. The former relative to daily durations is modifying daily observations according to projected anomalies (Damiano & Mercogliano 2013) or simulated data through statistical bias correction approaches (adopted for the Cervinara and Nocera Inferiore test cases). In the latter case, a stochastic approach is coupled with bias-corrected climate data to provide assessments at hourly scale (adopted for the Ravello test case)."*

At page 2, line 27 about the sentence "Moreover, in some studies (Reder et al. 2016; Ciervo et al. 2016; Rianna et al. 2017a, 2017b), slope stability conditions are assessed through expeditious statistical approaches referring to rainfall thresholds, while physically based approaches are preferred in other cases": which is the link with the sentences at lines 24-27? Moreover, it is also hard to understand. Perhaps the authors mean that critical rainfall thresholds for land slide in some cases are derived on the base of statistics of ??????? and in some cases through slope stability analysis? If this is the case, this is not clear.

At page 2, line 29 the sentence "Finally, climate projections at 8km in the optimized configuration over Italy (Bucchignani et al. 2015) and the Zollo et al. (2014) configuration of COSMO_CLM model (the highest resolution currently available for Italy up to 2100) are used as inputs in the aforementioned case studies, while climate projections from the Euro-CORDEX multimodel ensemble (Giorgi et al. 2016) are adopted in Rianna et al. (2017b). " could be rewritten specifying

that previous studies approached climate projections using two different methodologies. The former …. and the latter………………

*Response: The text has been significantly restructured according reviewer's remarks as: "Some studies (Reder et al. 2016; Ciervo et al. 2016; Rianna et al. 2017a), make use of expeditious statistical approaches referring to rainfall thresholds to assess slope stability conditions, while other studies employ physically based approaches (Rianna et al., 2017b). Moreover, weather input for impact models provided by climate projections at 8km in the optimized configuration over Italy of COSMO_CLM model, the highest resolution currently available for Italy up to 2100, (Bucchignani et al. 2015) are used as inputs in the all aforementioned case studies, except Rianna et al. (2017b) where uncertainties in climate projections are accounted for through Euro-CORDEX multimodel ensemble (Giorgi et al. 2016)."*

At page 3, line 3 the sentence "The present study focuses again on the Nocera Inferiore site, and also makes use, as will be discussed, of rainfall data from the sites of Gragnano and Castellammare di Stabia" could be simply rewritten as "The present study focuses on the Nocera Inferiore site, with the use of rainfall data from the rain gauges located in Gragnano and Castellammare di Stabia"

At page 3, line 6 the writer suggests to introduce the object of the study. The elements of novelty should be written after it.

*Response: The paragraph has been significantly restructured to accommodate the Reviewer's suggestion.*

**REVIEWER 2**

In abstract, authors mention "what they did" but never present "what they found".

Chapter on the calculation of the landslide hazard was newly added in the manuscript (new chapter 7). Because just calculation results are shown in the chapter, readers cannot know how to understand (and evaluate) the results. Some discussion on the landslide hazard is expected.

***Response:*** *We wish to thank the Reviewer for the remarks. The abstract has been thoroughly reviewed according to the Reviewer's suggestions and to maintain the number of words within the expected threshold (200). Furthermore, some insights about hazard have been added in Section 7.*

[revised manuscript text omitted]

---

## Author Response (AR3)

**Response to reviewers**

**Reviewer 1**

1) The authors improved the paper. However, there is a point to be exploited yet. The hazard they estimate, results from a probability calculation. Really, according to the Technical Italian Law, the estimation of hazard is quantitative and is usually obtained after considering three scenarios of different return period. The hazard the authors determine, seems a susceptibility hazard rather than a hazard. Therefore, the authors in the introduction should defining the hazard they are investigating after introducing the other hazard approaches.

*Response: The text has been amended to include explicit references to leading contributions in landslide risk glossary definitions including Corominas et al. (2005) and Fell et al. (2008) and to explain the conformity of the adopted approach with these glossaries.*

2) Authors should also add that the "quantitative hazard" is determined in different way by using models able to simulate both deposition and entrainment (Deangeli, 2008, Rosatti and Begnudelli, 2013; Frank et al. 2015, Stancanelli et al. 2015, Cuomo et al. 2016, and Gregoretti et al. 2018).

*Response: The text has been amended to include explicit references to these notable literature contributions.*

The following are the detailed comments to the unclear sentences.

At page 1 line 20, what does it mean "concentration scenario"?
*Response: The text has been amended to make clearer the last sentence*

At page 1 line 25, while (conversely)????? The authors should choose one.
*Response: The text has been amended as suggested.*

At page 1 line 25 there are two commas
*Response: The text has been amended as suggested.*

At page 2, lines 21-22 not clear sentence
*Response:The text has been modified in attempt to improve the form: "Indeed, at some locations (Cervinara and Nocera Inferiore), the triggering event is recognized to be characterized by daily duration (also jointly acting with wet antecedent conditions), while in other locations (Ravello) events in cover shallower or formed by coarser material require heavy rainfall lasting few hours.*

At page 2 line 27: references missing for the other studies
*Response: We thank the Reviewer; they have been added*

At page 3 line 7: The meaning of the sentence "Flow-like landslide…..." does not match the manuscript: the reach probability is the second step after the triggering probability for estimating the hazard. It is not an added value, something more as the sentence shows.
*Response: To stress the relevance of this stage, the sentence is moved above.*

At page 3, line 25: what is it ZOB? Please define it.

*Response: The definition "zero order basin" has been added to the text*

At page 3 lines 24-27: the difference between channelized and un-channelized is not clear.
***Response:***
*We added a clarification in brackets. The propagation process of a "channelized debris flows" is laterally confined by the stream channel; the propagation process of a "un-channelized debris flows" is not laterally confined.*

At page 4, line 2 "….estimate of hazard and its mapping" is better
***Response:*** *The text has been amended as suggested.*

At page 4 the sentence "The study area was modelled into the GIS software through a digital terrain model (DTM) having a resolution of 15x15 m." is rather unclear.

***Response:*** *The text has been amended for sake of improved clarity.*

[revised manuscript text omitted]